# Structural basis of MICAL autoinhibition

Matej Horvath [1,2], Adam Schrofel[1,2], Karolina Kowalska [1,2], Jan Sabo [3], Jonas Vlasak [1,2], Farahdokht Nourisanami[1,2], Margarita Sobol[1,2], Daniel Pinkas[4], Krystof Knapp [1,2], Nicola Koupilova[1,2], Jiri Novacek[4], Vaclav Veverka [1,5], Zdenek Lansky [3] & Daniel Rozbesky [1,2] ✉

MICAL proteins play a crucial role in cellular dynamics by binding and disassembling actin filaments, impacting processes like axon guidance, cytokinesis, and cell morphology. Their cellular activity is tightly controlled, as dysregulation can lead to detrimental effects on cellular morphology. Although previous studies have suggested that MICALs are autoinhibited, and require Rab proteins to become active, the detailed molecular mechanisms remained unclear. Here, we report the cryo-EM structure of human MICAL1 at a nominal resolution of 3.1 Å. Structural analyses, alongside biochemical and functional studies, show that MICAL1 autoinhibition is mediated by an intramolecular interaction between its N-terminal catalytic and C-terminal coiled-coil domains, blocking F-actin interaction. Moreover, we demonstrate that allosteric changes in the coiled-coil domain and the binding of the tripartite assembly of CH-L2α1-LIM domains to the coiled-coil domain are crucial for MICAL activation and autoinhibition. These mechanisms appear to be evolutionarily conserved, suggesting a potential universality across the MICAL family.

MICALs are a phylogenetically conserved family of actin regulatory enzymes that bind and disassemble actin filaments (F-actin)[1–3]. Unlike well-known actin disassembly proteins like gelsolin and cofilin, which act through conformational changes, MICALs utilize a unique oxidation-based mechanism. In particular, MICALs bind directly to F-actin and stereospecifically oxidize two conserved methionine residues (Met44 and Met47) into methionine sulfoxides[3,4]. These residues are located in the D-loop of actin, a critical region for the formation of longitudinal contacts between actin subunits[5,6]. The oxidation of these methionines has been shown to induce conformational changes in the D-loop, leading to F-actin destabilization[7]. The oxidative modification induced by MICAL on actin is reversible, with a class of enzymes known as methionine sulfoxide reductases MsrB/SelR able to restore the original state of the methionines[4,8].

MICALs have emerged as key players, orchestrating a myriad of cellular processes requiring discrete changes in the cytoskeleton. Their role is particularly pronounced in the nervous system, where they are essential for axon guidance[1], the regulation of growth cone morphology[9,10], the formation of neuronal connections[11,12], and modulation of neuronal migration[13]. Beyond their neural functions, MICAL proteins also significantly influence muscle organization[12,14,15], cardiovascular dynamics[16,17], and angiogenesis[18–20]. They are central to various cellular processes including specifying cell morphology, regulating migration, proliferation, cytokinesis, and vesicle trafficking (reviewed in ref. 21). Conversely, MICAL malfunctions have been linked to various pathologies, including cancer and neurological disorders like neurodegeneration (reviewed in ref. 21), underscoring their potential as therapeutic targets.

MICALs are large, multidomain intracellular proteins. Their catalytic activity is linked to an N-terminal monooxygenase domain (MO) that spans roughly 500 residues[22,23]. The MO non-covalently binds the FAD cofactor and uses NADPH and oxygen to drive its oxidative mechanisms. The MO is followed by three smaller domains: a Calponin-homology (CH) domain[24,25], a LIM domain, and a C-terminal

[1]Department of Cell Biology, Faculty of Science, Charles University, Prague, Czechia. [2]Institute of Molecular Genetics of the Czech Academy of Sciences, Prague, Czechia. [3]Institute of Biotechnology of the Czech Academy of Sciences, Prague, Czechia. [4]Central European Institute of Technology, Masaryk University, Brno, Czechia. [5]Institute of Organic Chemistry and Biochemistry of the Czech Academy of Sciences, Prague, Czechia. ✉e-mail: rozbesky@natur.cuni.cz

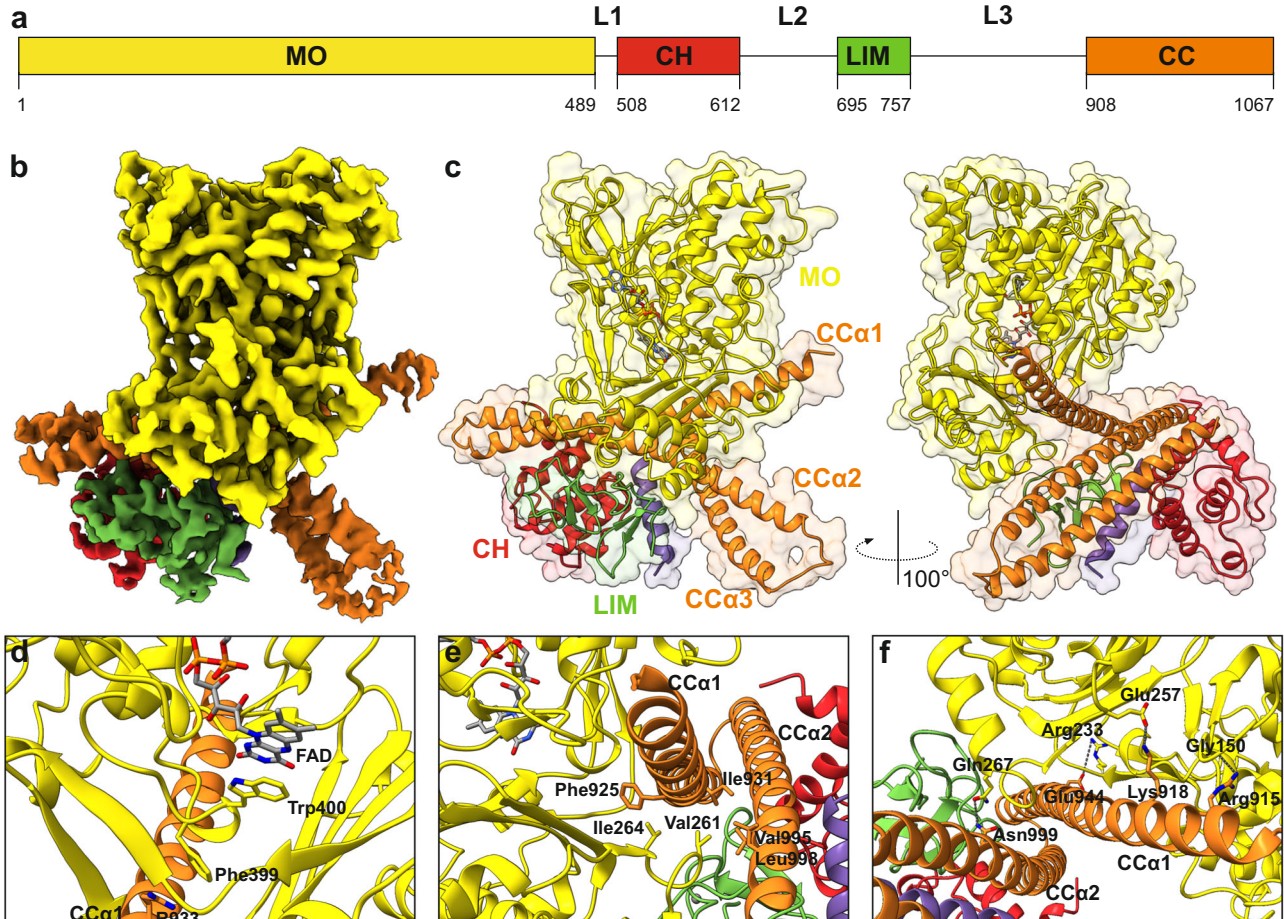

**Fig. 1 | Cryo-EM structure of human MICAL1. a** Schematic of the domain organization in human MICAL1, highlighting the monooxygenase (MO) domain, calponin homology (CH) domain, Lin-11, Isl-1, Mec-3 (LIM) domain, and coiled-coil (CC) domain, along with linker regions L1, L2, and L3. **b** Cryo-EM map of human MICAL1 with individual domains distinctly colored as shown in panel (**a**). **c** Ribbon representation of human MICAL1. **d** Detailed view of the region between MOβ17 and MOα15, showing key residues Trp400 and Phe399. Trp400 stabilizes the isoalloxazine ring of FAD via coaxial stacking, while Phe399 stabilizes the MO-CC interaction through a cation-π interaction with Arg933 from the CCα1 helix. **e** Hydrophobic contacts stabilizing the MO-CC interaction. Hydrophobic residues Val261 and Ile264, located between MOβ11 and MOα11, interact with a cluster of hydrophobic residues—Phe925, Val995, Leu998, and Ile931—from the CCα1 and CCα2 helices. **f** Polar contacts stabilizing the MO-CC interaction, primarily consisting of salt bridges (Arg233-Glu944, Glu257-Lys918) and hydrogen bonds (Gln267-Asn999, Gly150-Arg915).

coiled-coil (CC) domain[26,27]. These domains are interconnected by relatively long and flexible linker regions, which are known to mediate interactions with other proteins such as the adapter protein CasL[28], the non-receptor tyrosine kinase Abl[29] or an actin-based motor Myosin Va[30]. Although the structures of the individual domains have been described, the complete architecture of the full-length MICAL protein has yet to be revealed.

The catalytic activity of MICALs within cells must be tightly regulated, as imbalances in MICAL activity can lead to detrimental impacts on cellular morphology, primarily through the excessive disassembly of actin[2,31,32]. Previous studies have indicated that MICALs exist in an autoinhibited state[27,31,33], a condition from which they can be released through interactions with the cytoplasmic domains of plexin receptors[1,33] or with Rab proteins[26,34,35], a family of small GTPases. Yet, the precise molecular mechanisms underlying MICAL autoinhibition and activation remain largely elusive.

Here, we determined the cryo-EM structure of the full-length human MICAL1. Our findings, supported by biochemical and functional analyzes, indicate that MICAL1 autoinhibition hinges on the binding of the CH and LIM domains, facilitated by a helical region of the long linker, to the CC domain. This interaction maintains the CC domain in a specific conformation, enabling it to interact with the

catalytic domain and thus enforce autoinhibition. Conversely, we propose that during Rab-induced MICAL1 activation, Rab binding to the CC domain triggers the dissociation of the tripartite assembly, inducing allosteric changes in the CC domain that destabilize the autoinhibitory MO-CC interaction.

## Results

### Overall architecture of human MICAL1

We determined the structure of the human MICAL1 using single-particle cryo-EM to a nominal resolution of 3.1 Å (Fig. 1a–c, Figs. S1–2). The cryo-EM density map was clearly defined for all four domains but less visible for the linker regions, which were not built into the model (Fig. 1b). Central to the MICAL1 structure is its N-terminal monooxygenase (MO) domain (residues 1–489) with a high level of similarity to that of single mouse MO domain[22] with an rmsd of 0.69 Å over 479 Cα atoms. Analogous to its murine counterpart, the human MO domain comprises two subdomains of distinct sizes interconnected by two beta-strands (MOβ9 and MOβ15). A deep cavity located between these subdomains is partially occupied by the FAD cofactor, which, in the cryo-EM structure, is observed in its oxidized state with the isoalloxazine ring in a planar conformation.

The N-terminal MO domain interacts with the C-terminal coiled-coil (CC) domain (residues 908-1067) which is in agreement with the previous functional reports[27,31,33] suggesting a model of autoinhibition, in which the CC domain inhibits the catalytic function of MO. Similar to the previously published CC structures[26,27], the CC domain comprises three antiparallel helices (CCα1, CCα2, and CCα3) arranged in a coiled-coil motif. Central to the CC domain is helix CCα2, which wraps around helix CCα1 proximally and helix CCα3 distally. The interaction between the MO and CC domains is primarily mediated by the CCα1 helix, wedged within a shallow cavity between the large and small sub-domains of the MO (Fig. 1c). This shallow cavity is situated across from the deep cavity that houses the FAD cofactor. The interface between the MO and CC domains is extensive, burying a total solvent-accessible area of 2554 Å$^2$. The interaction interface is formed by a mixture of hydrophobic and hydrophilic interactions.

A key feature of the MO-CC interaction involves a region between MOβ17 and MOα15 (Fig. 1d, Fig. S1), potentially acting as a molecular switch. This region includes Trp400, which stabilizes the FAD's iso-alloxazine ring through coaxial stacking, and adjacent Phe399, stabilizing the MO-CC interaction via a cation-π interaction with Arg933 from the CCα1 helix (Fig. 1d). Another notable region that stabilizes the MO-CC interaction is between MOβ11 and MOα11 and contains hydrophobic clusters formed mainly by Val261 and Ile264, which form hydrophobic contacts with the cluster of hydrophobic residues Phe925, Val995, Leu998, and Ile931 from the CCα1 and CCα2 helices (Fig. 1e, Fig. S1). Additionally, other significant interactions encompass salt bridges (e.g., Arg233-Glu944 or Glu257-Lys918) and hydrogen bonds (e.g., Gln267-Asn999 or Gly150-Arg915) (Fig. 1f). Notably, the hydrogen bond involving Gly150-Arg915 is of particular interest, as previous studies have identified substitutions at these residues—specifically, Gly150Ser[36] and Arg915Cys[37]—in patients with Autosomal-Dominant Lateral Temporal Epilepsy (ADLTE). Intriguingly, the majority of these residues are highly conserved across sequences (Fig. S3).

The CH domain (residues 508-612) comprises a globular structure formed by six alpha-helices, stabilized by hydrophobic interactions within the core of the domain. This domain is highly similar to that of the CH domain of human MICAL1 determined via NMR[38]. In the cryo-EM structure, the CH domain binds to the CC domain. No apparent interactions were observed between the CH and MO domains, which is in contrast with prior crystallographic studies that reported contacts between the CH and MO domains[24,25]. However, these previous studies were based on a construct limited to the MO and CH domains. In the cryo-EM structure of MICAL1, the CH domain primarily interacts with the proximal part CCα2 helix through hydrophobic interactions (Fig. S4a).

Consistent with earlier predictions, the LIM domain (residues 695-757) features two contiguous zinc fingers, separated by a Phe-Arg pair (Fig. S4b). The LIM domain engages the central part of the CCα1 and CCα2 mainly through hydrophobic contacts (Fig. S4a). It also binds to the CH domain, with hydrophobic interactions between Phe716 from LIM and Val585 and Val586 from CH, further cemented by a salt bridge (Asp596 - Arg709). Intriguingly, the LIM domain also forms van der Waals contacts with the region between MOβ11 and MOβ12, incorporating MOα11, crucial for the stability of the MO-CC domain interaction (Fig. S4c).

## L2α1 helix from the L2 linker region stabilizes the CH-LIM-CC assembly

The four domains of MICAL1 are interconnected by three linker regions. The shortest linker, L1 (residues 490-507), serves as a bridge between the MO and CH domains. The intermediate L2 linker (residues 613-694) connects the CH domain to the LIM domain, and the longest, L3 (residues 758 to 907), links the LIM to the CC domain (Fig. 1a). The cryo-EM density for these linker regions was sparse, indicating their inherent flexibility and making it difficult to accurately fit. Despite this, we detected additional strong cryo-EM density in areas close to the main domains, suggesting that these densities may originate from the linker regions. Specifically, one of these densities appeared as a helical structure between the CH, CC and LIM domains, while another bulk density was observed near the proximal part of CCα1 (Fig. S5a, b). To better understand these extra densities, we used AlphaFold modeling of the linker regions followed by conducting 100 ns of molecular dynamics simulations (Fig. S5c−e). These simulations delineated the MO, CH, LIM, and CC domains as relatively rigid entities, with a few localized regions showing higher flexibility, notably observed in loop regions connecting helices in the CC domain (Fig. 2a, Fig. S5f). This pronounced flexibility in the CC domain loops may be important for maintaining the conformational

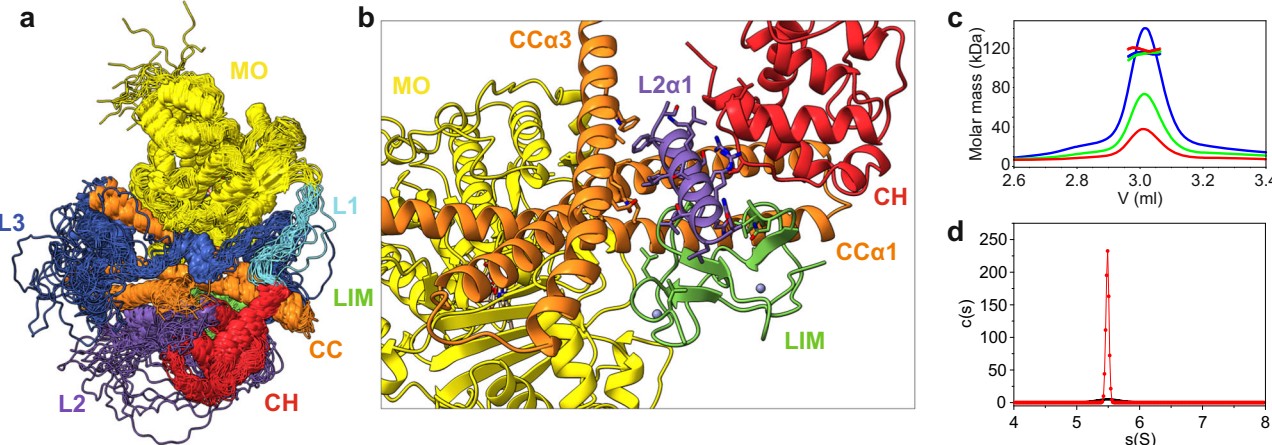

**Fig. 2 | Flexible linkers and their role in MICAL1. a** Molecular dynamics simulations of the cryo-EM structure of MICAL1, with flexible linkers modeled using AlphaFold. This representation illustrates the superposition of 21 backbone-traced conformers extracted at 5 ns intervals during molecular dynamics simulations, highlighting the dynamic nature of the linkers. **b** The amphipathic L2α1 helix, derived from the L2 linker region, is wedged between the CC, CH, and LIM domains. This helix is essential for the stability of the CH-LIM-CC assembly. **c** Molecular mass determination of MICAL1 in solution via size-exclusion chromatography combined with multi-angle light scattering (MALS), showing an experimental molar mass of 115.1 ± 3.6 kDa. This result corresponds with the theoretical molar mass for a monomer (124.8 kDa), with no peak shift towards higher molecular masses. The initial protein concentrations of 2.0 (blue), 1.0 (green), and 0.5 mg/ml (red) were used. **d** Sedimentation coefficient distribution of MICAL1 as determined by sedimentation velocity analytical ultracentrifugation at concentrations of 40 μM (red) and 8 μM (black). The calculated sedimentation coefficient (sw(20,w) = 5.5 S) is consistent with the expected value for a monomer.

freedom of the entire CC domain, allowing for adjustments in the relative positioning of its helices.

In contrast, the linker regions (L1, L2, and L3) demonstrated significant flexibility, exploring a wide range of conformations—from extended, flexible structures to more compact, helical arrangements (Fig. S5f). Notably, one of these helical conformations was associated with the CH, LIM, and CC domains, corresponding to the additional cryo-EM densities. Given the high quality of the cryo-EM density in this region, we confidently modeled this helix within the L2 linker (residues 630–647), which we have labeled L2α1. This helix is wedged between the CC, CH, and LIM domains and plays a crucial role in the stability of the CH-LIM-CC assembly (Fig. 2b). Additionally, the L2α1 helix demonstrated relative rigidity in molecular dynamic simulations (Fig. S5g,h). The L2α1 helix displays an amphipathic nature, with a polar face enriched in positively charged residues and a hydrophobic face engaging in interactions with hydrophobic clusters located centrally in CCα2, CCα3 (Fig. S4e), as well as the second zinc finger motif of the LIM domain and the CHα4-CHα5 region. Intriguingly, most of these residues are evolutionarily conserved (Fig. S3). In contrast, the extra density near the proximal part of CCα1 was not as well-defined as that of the L2α1 helix (Fig. S5b). This region likely represents an arrangement of small helices, with a narrow density extending from it, suggesting the presence of a peptide chain. While AlphaFold predicted this region to be proline-rich, the low local resolution hindered our ability to model this segment unambiguously or confirm it experimentally.

Considering the elongated and flexible nature of the linkers, we explored the potential of MICAL1 to undergo dimerization or oligomerization through domain swapping mechanisms. Analyzes conducted using size-exclusion chromatography coupled with multi-angle light scattering (SEC-MALS) revealed MICAL1 as a monomer (Fig. 2c), with no evidence supporting oligomerization. This observation was further supported by sedimentation velocity experiments designed to assess oligomeric states at increased protein concentrations, which likewise did not detect any signs of oligomerization up to a concentration of 40 μM (Fig. 2d). These findings align with prior reports[39], confirming MICAL1 is a monomer in solution.

## Binding of the CH-L2α1-LIM assembly to the CC domain is crucial for MICAL autoinhibition

Previous studies have highlighted the CC domain as crucial for MICAL1's autoinhibition[27,31,33]. We explored the role of the CC domain in F-actin depolymerization using a pyrene-labeled F-actin depolymerization assay alongside real-time total internal reflection fluorescence (TIRF) microscopy to visualize individual actin filaments and determine their depolymerization rate. Consistent with the autoinhibition hypothesis, the addition of purified MICAL1 to actin filaments did not affect depolymerization rates compared to controls (Fig. 3a, b, Fig. S6). However, depolymerization assays with the purified MO domain showed a significant increase in activity, with depolymerization rates of $11.6 \pm 4.9$ subunits/s, considerably faster than the control filaments ($1.78 \pm 0.9$ subunits/s) (Fig. 3a, c, Fig. S6). We then purified the CC domain and confirmed its helical fold using circular dichroism (CD) spectroscopy (Fig. S7). Surprisingly, the addition of the purified CC domain at varying concentrations to the MO failed to inhibit depolymerization (Fig. 3a, c, Fig. S6). Subsequent Biolayer interferometry (BLI) binding experiments between the MO and CC domains showed no measurable indications of binding up to an MO concentration of 27 μM (Fig. 3d). However, repeating BLI experiments with a truncated MICAL$^{\Delta CC}$ construct (lacking the CC domain) showed binding between the purified CC domain and MICAL$^{\Delta CC}$, with a dissociation constant ($K_D$) of $0.58 \pm 0.17$ μM (Fig. 3d). This construct depolymerized actin filaments at a slightly reduced rate ($5.51 \pm 2.73$ subunits/s) compared to the MO domain alone (Fig. 3a, e, Fig. S6). Notably, the addition of the purified CC domain to MICAL1$^{\Delta CC}$ led to a noticeable inhibition of F-actin depolymerization (Fig. 3a, e, Fig. S6). Data from the pyrene-labeled actin

depolymerization assays (Fig. 3e) were consistent with the trends observed in the single actin filament TIRF microscopy (Fig. 3a, Fig. S6), though the degree of inhibition observed in the pyrene assays was less pronounced. Interestingly, even at CC domain concentrations near saturation (94.4% bound at 10 μM, as calculated based on the determined $K_D$), the purified CC domain did not inhibit MICAL1$^{\Delta CC}$ to the same extent as full-length MICAL1. This discrepancy could be attributed to multiple CC conformations in solution or an overestimation of the $K_D$ in the BLI experiment due to the absence of F-actin. We then purified the MICAL1$^{\Delta MO}$ construct (lacking the MO domain) to determine whether it could inhibit the MO domain's activity using the pyrene-labeled F-actin depolymerization assay. The addition of MICAL1$^{\Delta MO}$ to the purified MO domain resulted in substantial inhibition of the MO domain's depolymerization activity across three different molar ratios, with the level of inhibition comparable to that observed with full-length MICAL1 (Fig. 3f). These findings suggest the purified CC domain alone cannot autoinhibit the MO domain's depolymerization activity, indicating that autoinhibition requires additional elements from the L1-CH-L2-LIM-L3 region.

The structure of MICAL1 underscores the presence of the tripartite CH-L2α1-LIM assembly. Superimposition of the autoinhibited MICAL1 structure with the previously reported crystal structure of the CC domain complexed with Rab10, reveals several key insights. First, notable differences emerge in the conformation of the CC domain between the autoinhibited MICAL1 and the CC-Rab10 complex, particularly in the pitch angle of the CCα3 helix (Fig. 4a). While the CC domain in the Rab10 complex shows a more planar orientation, in autoinhibited MICAL1, CCα3 exhibits an axial tilt, disrupting this planarity due to its interaction with the L2α1 helix (Fig. 4b). To clearly differentiate these conformations, we will refer to the axially tilted conformation in autoinhibited MICAL1 as CC$^{inh}$ and the planar conformation in the CC-Rab10 complex as CC$^{act}$, when necessary. Second, the CC$^{act}$ low-affinity binding site for Rab10 in the complex overlaps with the CC$^{inh}$ binding site for the tripartite assembly CH-L2α1-LIM in autoinhibited MICAL1 (Fig. 4c), indicating mutually exclusive binding to the CC domain. Third, the axial tilt of CCα3$^{inh}$ in autoinhibited MICAL1 prevents Rab10 from binding to the high-affinity binding site on CC (Fig. 4c). Fourth, shifts in the orientation of the proximal part of CCα1 are also significant. The reduced crossing angle between CCα1$^{act}$ and CCα2$^{act}$ in the CC-Rab10 complex is responsible for a shift in proximal CCα1$^{act}$ from the MO domain when compared to its position in the autoinhibited state. This displacement compromises critical interactions that stabilize the MO-CC domain interface, including those involving residues Glu257-Lys918, Gly150-Arg915, and Gln267-Asn999 (Fig. 4d). Additionally, this leads to the compromised cation-π interaction Phe399-Arg933, which is adjacent to Trp400, responsible for stabilizing FAD through coaxial stacking (Fig. 4e). Indeed, when Arg933 was substituted with alanine, we observed a significant release of MICAL1 autoinhibition in the pyrene-labeled F-actin depolymerization assay (Fig. 3g).

In summary, our results highlight the pivotal role of the CH-L2α1-LIM tripartite assembly in enabling the CC domain to adopt the tilted conformation (CC$^{inh}$) required to bind and inhibit the MO domain's activity. This is supported by the observation that the MICAL1$^{\Delta MO}$ construct effectively inhibited the MO domain's depolymerization activity, with a degree of inhibition comparable to that of the full-length MICAL1. We propose that dissociation of the CH-L2α1-LIM assembly, potentially triggered by Rab protein binding, induces a conformational change that locks the CC domain in an active planar conformation (CC$^{act}$), preventing it from re-engaging in autoinhibition. This is consistent with our observation that the purified CC domain alone, which seems to adopt a planar conformation[27], was unable to inhibit the MO domain, and that its inhibition of MICAL1$^{\Delta CC}$ was less effective than that in the full-length protein. Together, these results indicate that autoinhibition of MICAL1 is not solely mediated by the CC domain but requires the cooperative involvement of the CH-L2α1-LIM assembly.

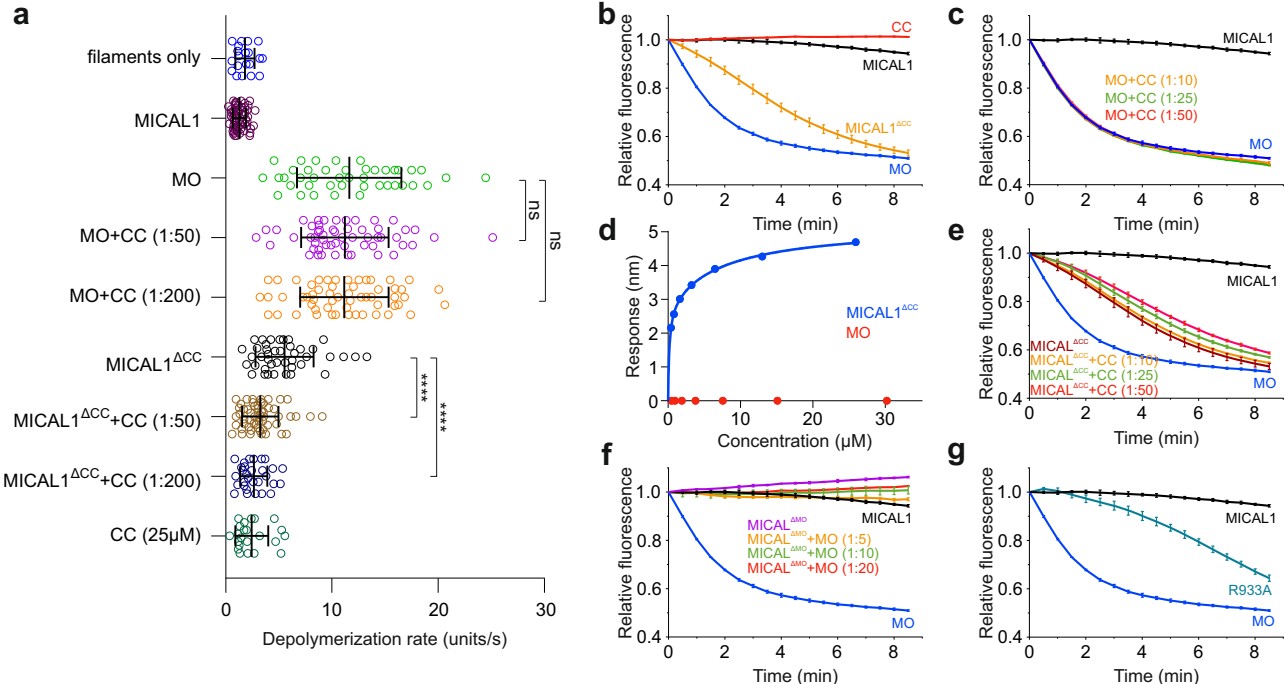

**Fig. 3 | MICAL1 autoinhibition relies on the binding of the CH-L2α1-LIM assembly to the CC domain. a** Single-actin filament TIRF microscopy data revealed that treatment of F-actin with purified full-length MICAL1 did not alter depolymerization rates (1.27 ± 0.61 units/s, $N = 60$) compared to the control (1.78 ± 0.9 units/s, $N = 20$). Conversely, the addition of the purified MO domain resulted in a significant increase in depolymerization rates (11.61 ± 4.91 units/s, $N = 42$). The addition of purified CC to the MO domain at molar ratios of MO to CC at 1:50 or 1:200 exhibited no measurable effect on the depolymerization rate (11.20 ± 4.12 units/s, $N = 55$ for the 1:50 ratio and 11.15 ± 4.16 units/s, $N = 57$ for the 1:200 ratio), suggesting that the CC domain alone does not inhibit the MO domain. MICAL1$^{\Delta CC}$ exhibited a lower rate of depolymerization (5.51 ± 2.73 units/s, $N = 42$) compared to that of the MO domain. However, the addition of purified CC to MICAL1$^{\Delta CC}$ significantly inhibited depolymerization rates (3.23 ± 1.71 units/s, $N = 56$ and 2.62 ± 1.23 units/s, $N = 30$ for molar ratios of 1:50 and 1:200, respectively). As a control, the addition of purified CC alone did not affect the depolymerization rate of F-actin (2.43 ± 1.55 units/s, $N = 23$) compared to untreated control filaments. In this experiment, the total concentrations were as follows: MICAL1, MO, and MICAL1$^{\Delta CC}$ at 500 nM; NADPH at 200 μM; and CC at final concentrations of 25 μM (molar ratio 1:50) and 100 μM (molar ratio 1:200). The estimated surface concentration of actin was 50 molecules/μm². Data are presented as mean values ± standard deviation of depolymerization rates of individually measured actin filaments ($N$). $P$-values were calculated using a nonparametric unpaired two-tailed t-test with Welch's correction: MO + CC (1:50), $p = 0.6678$ (ns); MO + CC (1:200), $p = 0.6281$ (ns); MICAL1$^{\Delta CC}$ + CC (1:50), $p < 0.0001$ (****); MICAL1$^{\Delta CC}$ + CC (1:200), p < 0.0001 (****); Representative micrographs are shown in Fig. S6. **b**, **c**, **e** Data

from pyrene-labeled actin depolymerization assays demonstrate overall trends that are consistent with the findings from TIRF microscopy (**a**). In this experiment, the concentrations were as follows: MICAL1, MO, and MICAL1$^{\Delta CC}$ at 200 nM; NADPH at 200 μM; CC at final concentrations of 2 μM (1:10 molar ratio), 5 μM (1:25 molar ratio), and 10 μM (1:50 molar ratio); and actin at a final concentration of 2 μM. Data are presented as mean values ± standard deviation of the three independent replicates ($n = 3$). **d** BLI binding experiments involved immobilization of biotinylated CC domain on sensor tips and analysis for binding with MO or MICAL1$^{\Delta CC}$. While no measurable binding was observed for MO up to a concentration of 27 μM, MICAL1$^{\Delta CC}$ exhibited specific binding with a $K_D$ of 0.58 ± 0.17 μM. The data were measured in independent duplicates. **f** Pyrene-labeled actin depolymerization assays showed that MICAL1$^{\Delta MO}$ significantly inhibited the depolymerization activity of the MO domain at all tested molar ratios, with inhibition levels similar to that observed with the full-length MICAL1. The concentrations used were as follows: MICAL1 and MO at 200 nM; NADPH at 200 μM; MICAL1$^{\Delta MO}$ at 1 μM (1:5 molar ratio), 2 μM (1:10 molar ratio), and 4 μM (1:20 molar ratio); and actin at 2 μM. Data are presented as mean values ± standard deviation of the three independent replicates ($n = 3$). **g** Pyrene-labeled actin depolymerization assays demonstrated that disrupting the interaction between the MO domain and the CCα1 helix releases MICAL1 autoinhibition. Specifically, substituting Arg933 in CCα1, which forms a cation-π interaction with Phe399 in the MO domain in autoinhibited MICAL1, resulted in significant F-actin depolymerization. The concentrations used were as follows: MICAL1, MO, and MICAL1$^{R933A}$ at 200 nM; NADPH at 200 μM; and actin at 2 μM. Data are presented as mean values ± standard deviation of the three independent replicates ($n = 3$).

## Overlap of the CCα1$^{inh}$ helix and F-Actin binding sites on the MO Domain

Previous studies have reported that the MO domain can bind F-actin[2,3,32,40,41]. To explore the potential competitive interaction between the CC$^{inh}$ domain and F-actin for binding to the MO domain, we conducted a Hydrogen–Deuterium Exchange Mass Spectrometry (HDX-MS) experiment. This technique offers detailed insights into the solvent accessibility and dynamic properties of proteins.

To prevent the depolymerization of F-actin by the MO domain, we omitted the NADPH cofactor from the samples during these experiments. In the HDX experiment, the MO domain was exposed to deuterated water ($D_2O$) in the presence and absence of F-actin. This exposure allowed the amide hydrogens in flexible or exposed regions of the MO domain to exchange with deuterium more readily than those in less exposed or F-actin buried regions (Fig. 5a). After specific time intervals, we quenched the exchange

and digested the deuterium-labeled proteins with proteases. The resulting peptides were then analyzed using mass spectrometry, and deuterium uptake levels were compared between MO peptides from samples with and without F-actin.

Our analysis revealed several regions with significantly reduced deuterium uptake in the MO-F-actin complex compared to the MO domain alone (Fig. 5b, Fig. S8, Supplementary Table 2). Among these, the region encompassing MOβ11, MOα11, and MOβ12, demonstrated the most pronounced decrease in deuterium uptake. This region, as indicated by the cryo-EM structure, interacts with both the LIM domain and the CCα1 and CCα2 helices (Fig. S4c). Another notable area of reduced deuterium uptake was found in the loop between MOα16 and MOβ18 (residues 440-453), which is positioned just above CCα1 in the cryo-EM structure. Furthermore, a significant reduction in deuteration was observed in the MOβ4 region and its adjacent loop (residues 148-158), which includes key residues such as Gly150 that interact with R915

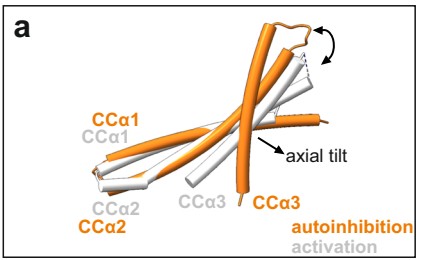
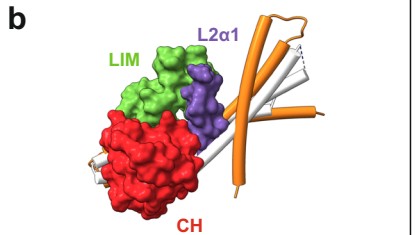
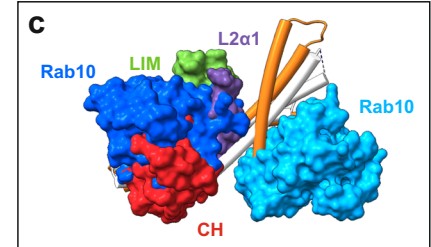

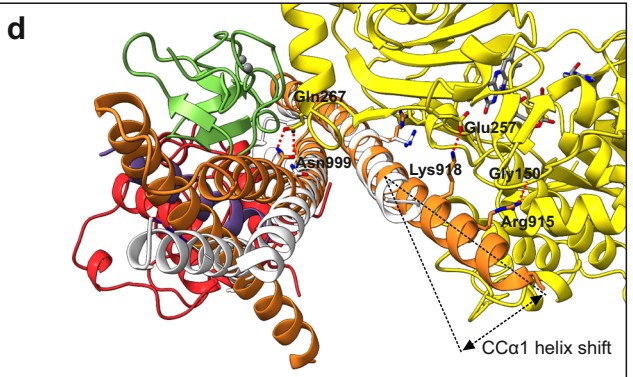
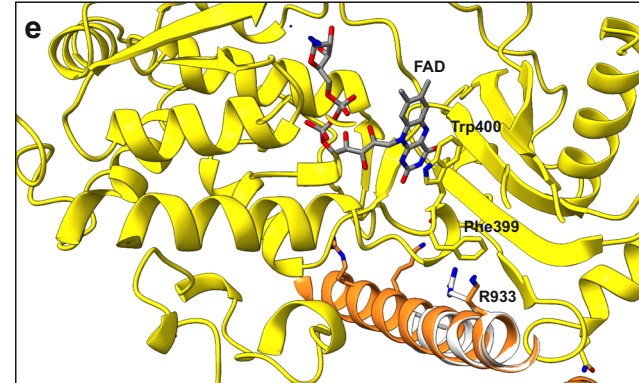

**Fig. 4 | Rab10-induced conformational changes in the CC domain decrease the MO-CC stability. a** Overlay of the CC domain in its autoinhibited (orange) and Rab-activated (white) states. The overlay was generated by superimposing the CC domains from the previously reported crystal structure of the CC-Rab10 complex[26] and the cryo-EM structure from the current study. In the activated state, the CC domain exhibits a more planar arrangement of individual helices, whereas in the autoinhibited state, the CCα3 helix displays an axial tilt, which disturbs the CC planar arrangement. **b** The same CC domains as shown in (**a**) overlaid with the CH-L2α1-LIM assembly from the cryo-EM structure. This overlay demonstrates that the L2α1 helix maintains the CCα3 helix in the axially tilted conformation. **c** Further overlay of the structures from (**b**) with two Rab10 molecules from the previously

reported CC-Rab10 complex[26]. The low-affinity binding site of Rab10 overlaps with the CH-L2α1-LIM assembly, whereas Rab10 bound to the high-affinity site clashes with the axially tilted CCα3 helix. **d** Close-up view illustrating the shift of the proximal region of the CCα1 helix between the Rab-activated and autoinhibited states. In the Rab-activated state, the helix shifts away from the MO domain, disrupting critical interactions that stabilize the CC binding to the MO domain. **e** Detailed view highlighting differences in residue Arg933 between the activated and autoinhibited states. In the autoinhibited state, Arg933 in CCα1 forms a cation-π interaction with Phe399 from the MO domain, which is adjacent to Trp400, stabilizing FAD via coaxial stacking. Conversely, in the Rab-activated state, this cation-π interaction between Arg933 and Phe399 is disrupted.

of CCα1. Additional regions with decreased deuterium uptake included the helical segments MOα4 and MOα17, which, although not directly contacting CCα1, are located on the same face of the MO domain as the CCα1 binding site. Interestingly, the opposite face of the MO domain, which contains the deep cavity housing the FAD cofactor, exhibited minimal changes in deuteration, except for a minor helical region spanning residues 201-206. These observations suggest that the binding site for the CCα1[inh] helix on the MO domain partially or fully overlaps with the F-actin binding site. This finding is further supported by the high conservation of the CCα1 binding site on the MO domain, in contrast to the relatively lower conservation of the face containing the FAD-binding cavity (Fig. 5c). Overall, our HDX-MS results indicate that F-actin competes with the CCα1 for binding to the MO domain.

## Discussion

The MICAL family of proteins plays a crucial role in regulating actin dynamics. These proteins bind selectively to actin filaments, oxidizing actin and disrupting filament structure, ultimately leading to disassembly. Dysregulation of MICAL activity results in excessive disassembly of actin filaments, which adversely affects cellular morphology. This presents a conundrum: how is MICAL activity tightly regulated within the cell? While previous research has suggested that MICAL exists in an autoinhibited state[27,31,33], with its CC domain forming intramolecular interactions with the MO domain to enforce autoinhibition, the exact molecular mechanisms governing MICAL autoinhibition and the contributions of other MICAL domains to this process remain unclear.

In this study, we present structural, biochemical, and functional analyzes to elucidate the molecular mechanisms underlying the

autoinhibition of MICAL1 and provide insight into its activation induced by Rab. Consistent with previous research findings[27,31,33], our investigation underscores the critical role of the CC domain in mediating autoinhibition of MICAL1. Specifically, we demonstrated that the CC[inh] domain engages the MO domain through the CCα1[inh] helix. Our HDX-MS experiments suggest that the binding site of the CCα1[inh] helix on the MO domain either fully or partially overlaps with the F-actin binding site. Therefore, we propose that steric hindrance resulting from MO-CCα1[inh] binding constitutes a primary mechanism of autoinhibition, preventing interaction with F-actin (Fig. 6).

Furthermore, our study identifies a tripartite complex involving the CH domain, L2α1 helix, and LIM domain, collectively interacting with the CC[inh] domain. Notably, the L2α1 helix plays a pivotal role by maintaining the CCα3[inh] helix in an axially tilted conformation, facilitating engagement of CCα1[inh] with the MO domain through a combination of hydrophobic and hydrophilic interactions. A critical interaction within this network involves a cation-π interaction between Arg933 of CCα1 and Phe399 within the MO domain. Adjacent to Phe399 lies Trp400, which stabilizes the FAD cofactor by coaxially stacking with its isoalloxazine ring, thus anchoring the FAD cofactor firmly within the deep cavity of the MO domain and likely hindering its reduction by NADPH. Therefore, inhibition of FAD cofactor reduction emerges as another significant autoinhibitory mechanism in MICAL1. These findings are in agreement with previous studies on MICAL1's NADPH oxidase activity, revealing a 200-fold decrease in catalytic efficiency ($k_{cat}/K_{NADPH}$) for full-length MICAL1 compared to the MO domain alone[42].

We also offer insights into the mechanism of Rab-induced MICAL1 activation (Fig. 6). In this model, the low-affinity binding site for Rab10

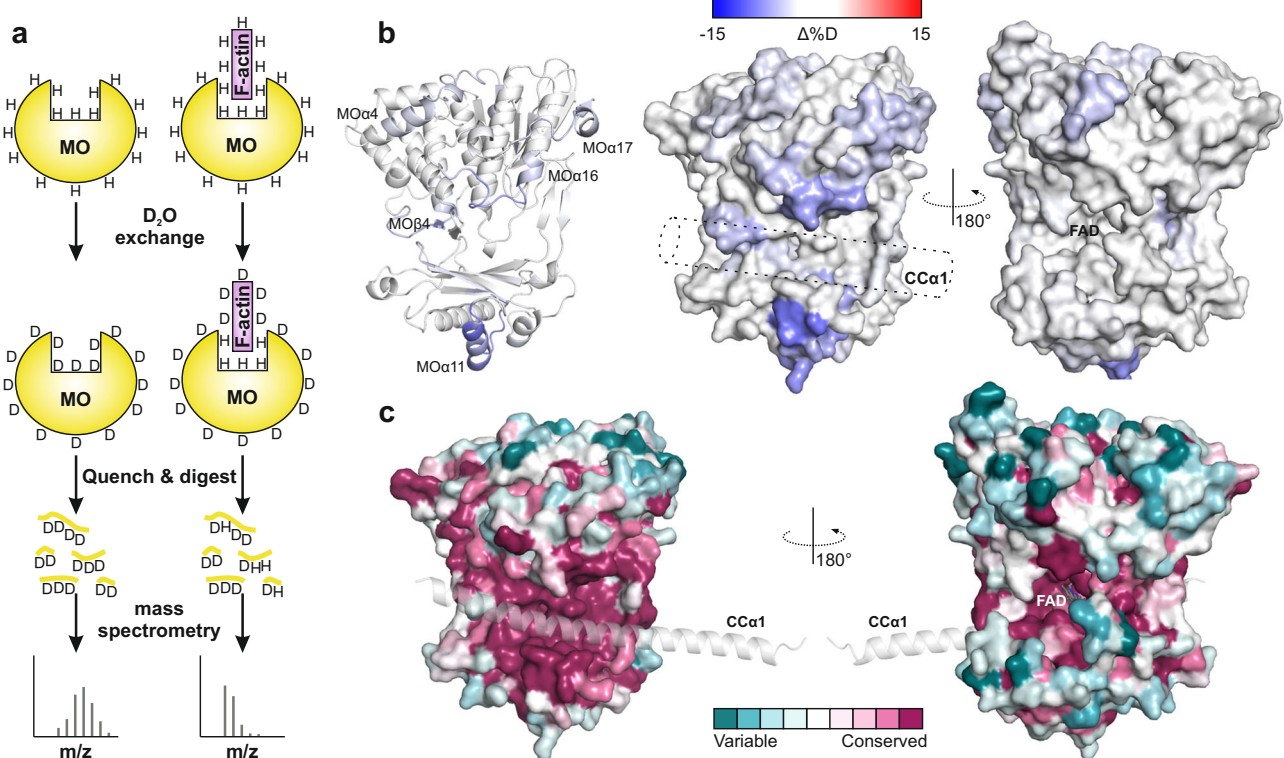

**Fig. 5 | F-actin and the CCα1 helix compete for binding to the MO domain.**
**a** Schematic representation of the HDX-MS experiment. The MO domain, both in the absence and presence of F-actin, was diluted into $D_2O$ buffer, allowing hydrogen-deuterium exchange over time. Exchange was quenched at various time points, after which the samples were digested, and the deuterium content was quantified using mass spectrometry. The following concentrations were used: 50 pmol of MO at 12.4 μM per reaction. The MO-F-actin complex was prepared by mixing MO and F-actin at a 1:2 molar ratio. **b** Local differences in amide hydrogen-deuterium exchange between the MO domain in the absence and presence of F-actin. These differences, measured after a 2-hour deuteration period, are mapped

onto both the ribbon and surface representations of the MO structure. Regions with the highest exchange levels are shown in red, while those with the lowest exchange are in blue. The position of CCα1 is schematically indicated by a dashed cylinder. **c** Visualization of the MO domain with CCα1 helix. The MO domain is presented in a surface representation with the CCα1 helix in a ribbon representation. Color coding reflects residue conservation from sequence alignments shown in Fig. S2, highlighting the most conserved areas near the CCα1 binding site. In contrast, areas surrounding the FAD-binding cavity show varied conservation levels, with direct FAD-contacting residues exhibiting significant conservation. The analyzes were conducted using the Consurf server[68].

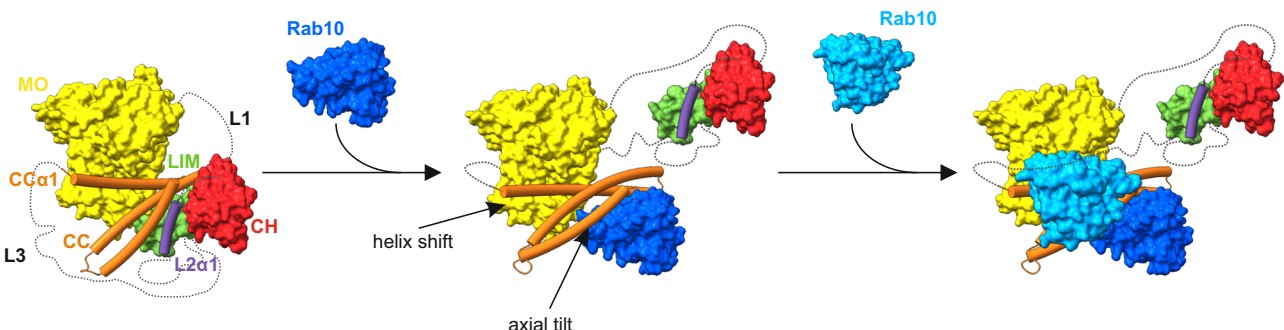

**Fig. 6 | Proposed Molecular Mechanism of MICAL1 Autoinhibition and Activation.** Initially, MICAL1 activity is autoinhibited by an intramolecular interaction between the MO domain and the CCα1^inh helix. This interaction sterically hinders F-actin from binding to the MO domain. A crucial component of this autoinhibitory mechanism is the tripartite CH-L2α1-LIM complex, which interacts with the CC^inh domain to maintain a CC conformation that facilitates the CCα1 helix's binding to the MO domain. The binding of Rab10 to MICAL1 initiates the dissociation of the

CH-L2α1-LIM complex, leading to the straightening or axial adjustment of the CCα3 helix. This triggers an allosteric change in the CCα1^inh helix, causing its proximal portion to shift away from the MO domain. While it remains uncertain whether this results in a complete or partial dissociation of the CC domain from the MO domain, this conformational shift likely exposes the F-actin binding site. Additionally, the conformation of the Rab-bound CC^act domain may be further stabilized by a second Rab molecule, enhancing the activation process.

on the CC^inh domain is initially occluded by the tripartite assembly CH-L2α1-LIM. Additionally, the axial tilt of CCα3^inh in the autoinhibited state of MICAL1 likely prevents Rab10 from binding to the high-affinity site on the CC domain. We propose, that initially, Rab10 binds to the low-affinity site within the CC domain, which disrupts the tripartite

assembly and leads to its dissociation from the CC domain. This dissociation removes the L2α1 helix, a key element maintaining the axial tilt of CCα3 in the autoinhibited state, thereby allowing CCα3 to straighten and enabling a transition of the entire CC domain to a more planar conformation. Once this planar conformation is achieved, a

second Rab10 molecule can bind to the newly exposed high-affinity site on the CC$^{act}$ domain. This second binding event is likely crucial for stabilizing the active conformation of the CC domain. The dual Rab binding may establish a more rigid and stable structural framework, preventing the CC domain from reverting to its autoinhibited state. This stabilization can be explained by allosteric effects, where Rab10 binding induces an energetically favorable conformational shift that locks the CC domain in its active state.

An alternative mechanism is also plausible, where the binding sequence of Rab10 is reversed. In this scenario, Rab10 could first bind to the high-affinity site, inducing the straightening of the CC domain and triggering the dissociation of the tripartite CH-L2α1-LIM assembly, thereby allowing a second Rab10 molecule to bind to the low-affinity site. Both binding sequences are theoretically feasible, and further studies, such as structural determination of MICAL1 in complex with Rab molecules, are necessary to resolve this ambiguity.

Concomitant with these conformational changes, the straightening of CCα3 induces an allosteric conformational change in CCα1. Specifically, the proximal portion of CCα1 undergoes a shift away from the MO domain, leading to the disruption of MO-CC interactions that stabilize the association between the CC and MO domains. It is not clear whether the helix shift results in a complete or partial dissociation of the CC domain from the MO domain. However, these changes likely result in the exposure of the binding site for F-actin and eliminate steric clashes that prevent F-actin binding to the MO domain. Additionally, the shift in CCα1 disrupts the cation-π interaction between Arg933 of CCα1 and Phe399 within the MO domain, leading to increased flexibility of FAD. This alteration is consistent with previous studies demonstrating that mouse MICAL-2 MO domain is reduced by NADPH approximately 100-fold faster in the presence of F-actin[40]. Ultimately, binding of F-actin to the MO domain, coupled with the reduction of the FAD cofactor, then leads to full activation of MICAL1. Notably, the MICAL1 cryo-EM structure revealed that one of the key autoinhibitory interactions between the MO and CC domains involves hydrogen bond between Gly150 and Arg915. These residues have recently been identified in patients with Autosomal-Dominant Lateral Temporal Epilepsy (ADLTE)[36,37], suggesting that disruption of this bond could release MICAL1 from autoinhibition and lead to pathological MICAL1 overactivation.

In this study, we present the molecular mechanisms governing MICAL1 autoinhibition and provide insight into its activation via Rab proteins. These mechanisms exhibit evolutionary conservation and are likely transferrable across the broader MICAL family, except for MICAL2, which lacks the CC domain and thus remains constitutively active. Further investigation will be required to tease out the precise roles of the CH and LIM domains in F-actin binding, as well as to explore the intricate molecular interplay between MICAL and F-actin, including the mechanisms underlying MICAL-induced F-actin depolymerization.

## Methods

### Protein production
Constructs encoding human MICAL1 (residues M1-G1067), MICAL1$^{\Delta MO}$ (residues A508-G1067) and MICAL1$^{\Delta CC}$ (residues M1-A917) were cloned into the pBacPAK9 vector (Takara) with an in-frame C-terminal 3C-His8 tag. A list of all oligonucleotides used in this study is shown in Supplementary Table 3. The proteins were then produced in baculovirus-infected Sf9 cells (ATCC CRL-1711). Three days post-infection, the Sf9 cells were harvested and subsequently lysed with a buffer comprising 30 mM Tris-HCl (pH 8.0), 500 mM NaCl, 20 mM imidazole, 3 mM DTT, 10 mM MgCl$_2$, 1% NP-40 (v/v), DNase (15 mg/L) and a protease inhibitor tablet (SigmaFast, Merck). The lysate was further processed by sonication and clarified through centrifugation at 40,000 g. For purification, the filtered lysate was subjected to immobilized metal-affinity chromatography using a HisTrap FF column 5 ml (Cytiva) and further

purified using size-exclusion chromatography with a Superdex 200 16/600 column (Cytiva).

The MICAL1$^{R933A}$ mutant was generated using site-directed mutagenesis with overlap-extension PCR. The purification process for the mutant protein followed the same protocol as for the wild-type, with one modification: the NaCl concentration in the lysis buffer was reduced to 300 mM for the mutant protein, compared to 500 mM for the wild-type. The mutant protein was expressed at levels similar to those of the wild-type.

Constructs encoding human MICAL1$^{MO}$ (residues T7-E489) and the CC domain (residues K918-G1067) were cloned into the pET45b (Merck) vector in-frame with a N-terminal His6-3C tag. MICAL1$^{MO}$ was produced in E. coli Rosetta2 cells at 15 °C in TB medium supplemented with 1 mM MgCl$_2$, 100 μM FeCl$_3$, and 0.1x BME Vitamin solution (MERCK). After 48 hours of induction with IPTG at a final concentration of 1 mM, the cells were harvested and subsequently lysed using a buffer containing 1xPBS, 1 M NaCl, 10 mM imidazole, 10 mM MgCl$_2$, 0.2% NP-40 (v/v), a protease inhibitor tablet (SigmaFast, Merck), 100 μM PMSF, 4 mM BME, lysozyme (1 mg/ml), and DNase (15 mg/L). The lysate was sonicated and clarified through centrifugation at 70,000 g. Purification involved immobilized metal-affinity chromatography with a 5 ml HisTrap Talon column (Cytiva), followed by cation exchange chromatography on a HiTrap SP 5 ml column (Cytiva). The final purification step employed size-exclusion chromatography on a Superdex 200 16/600 column (Cytiva).

The CC domain (residues K918-G1067) was produced in E. coli Rosetta2(DE3) cells. Following IPTG induction, the cells were incubated for 18 h at 18 °C. Afterward, the cells were harvested and lysed in a buffer consisting of 50 mM Tris-HCl (pH 8.0), 500 mM NaCl, 10 mM MgCl$_2$, 1% NP-40 (v/v), a protease inhibitor tablet (SigmaFast, Merck), lysozyme (1 mg/ml), and DNase (15 mg/L). The lysate was sonicated and clarified through centrifugation at 70,000 g. Subsequent purification involved immobilized metal-affinity chromatography using a 5 ml HisTrap Talon column (Cytiva) followed by size-exclusion chromatography with a Superdex 200 16/60 column (Cytiva).

### Cryo-EM sample preparation and data acquisition
Prior to grid plunging, purified MICAL1 in a buffer containing 15 mM Tris-HCl (pH 8.0), 150 mM NaCl, and 2 mM DTT was mixed with CHAPSO to reach a final CHAPSO concentration of 3 mM. After incubating with CHAPSO for 20 minutes, 4 μL of MICAL1 at a concentration of 3 mg/ml was applied to 300 mesh UltrAuFoil grids R 1.2/1.3. Following a 10-second incubation, the grids were blotted for 4.0 s and rapidly frozen in liquid ethane. Grid preparation was carried out using a Vitrobot Mark IV (Thermo Fisher Scientific) at 4 °C with humidity maintained between 90 and 100%.

Single particle cryo-EM data were collected in an automated manner on a Titan Krios G1 (Thermo Scientific) TEM, operated at 300 kV, using SerialEM software[43]. The microscope was aligned for fringe-free imaging and was equipped with a Bioquantum K3 (Ametek) direct electron detector. The camera operated in electron counting mode, and the data were collected at a pixel size of 0.834 Å/px. The microscope's condenser system was set to produce an electron flux of 21 e/Å$^2$s on the specimen, and data from 2.5–3.0 s exposures were stored into 40 frames. The energy-selecting slit was set to 20 eV. Data from 3×3 neighboring holes were collected using beam/image shifting while compensating for the additional coma aberration. The data were collected with the nominal defocus range from −1.4 to −2.8 μm. Overall, the dataset was composed of 59,961 movies. The data acquisition parameters are summarized in Supplementary Table 1.

### Cryo-EM data processing
All images were processed using CryoSparc v3.1.1[44] and Relion5[45]. Initially, 59,961 movies were imported into CryoSparc and corrected for the motion acquired during data acquisition using the Patch

Motion Correction job. The contrast transfer function parameters were estimated using the Patch CTF job. Micrographs were visually inspected, and those with poor-quality CTF fit, excessive aggregation, or ice artifacts were excluded from the dataset. The initial particle set was selected using the Blob picker with a particle size range of 100–130 Å, resulting in 6,256,757 particles. Following visual inspection, particles were extracted with a box size of 256 pixels. The extracted particles underwent four runs of 2D classification to remove false positives and corrupted particles, yielding 2,476,174 particles. These particles were subjected to multiple runs of Ab initio reconstruction, followed by heterogeneous refinement. From this process, 1,277,125 particles were selected and refined using the Non-uniform refinement job in CryoSparc. The particles were then imported into Relion for iterative processing which comprised 3D refinement, local CTF refinement, and Bayesian polishing. Subsequently, 3D classification without optimization of particle rotational parameters and shifts was used to split the particles into four classes. A class comprising 84,863 particles was selected from the 3D classification and subjected to yet another round of 3D refinement, CTF refinement, Bayesian polishing, and 3D refinement resulting in a 3.1 Å structure (FSC = 0.143) calculated from 84,863 particles.

## Model building, refinement, validation and analysis

The crystal structure of the mouse MICAL1 MO domain[22], mouse MICAL1 CH domain[24], human MICAL1 CC domain[27], and AlphaFold model of the LIM domain (AF-Q8TDZ2-F1) were manually fitted into the initial cryo-EM density map using UCSF Chimera[46]. One cycle of rigid-body real-space refinement was initially executed, followed by manual adjustments in Coot[47]. Subsequently, iterative rounds of Phenix[48] real-space refinement, coupled with manual building in Coot[47], were applied to enhance the overall model geometry. Validation of the cryo-EM structure was performed using Phenix comprehensive validation tools[48] and MolProbity[49]. Validation metrics included Fourier Shell Correlation (FSC) between half-maps and model-map comparisons, Real Space Cross-Correlation (RSCC), geometric quality assessments (Ramachandran plots, bond lengths, and angles), and overall clashscore. These metrics confirmed a high-quality model with minimal overfitting and an accurate representation of the experimental data. Key validation metrics are shown in Fig. S3, and the map and model statistics are detailed in Supplementary Table 1. Local resolution estimation of the cryo-EM map was computed using Phenix[48], while the buried surface areas of protein–protein interactions were determined with PISA[50]. The RMSD between aligned pairs of the backbone C-alpha atoms in superposed structures was calculated using PDBeFold[51], and sequence alignments were generated with KAlign[52]. Figures were created using UCSF Chimera[46], ESPRIPT,[53] and Corel Draw (Corel Corporation).

## Size-exclusion chromatography with multi-angle light scattering

MICAL1 was subjected to size-exclusion chromatography on an Agilent Biosec-3 column (4.6 × 300 mm) with a flow rate of 0.3 ml/min, using a mobile phase consisting of 15 mM Tris-HCl (pH 8.0), 150 mM NaCl and 2 mM DTT at a temperature of 25 °C. The SEC column was coupled with a suite of detectors, including a miniDAWN static light-scattering detector (Wyatt Technology), a Shodex RI-501 differential refractive index detector, and an Agilent 1260 Infinity II UV detector (Agilent Technologies). The acquired data were analyzed using the ASTRA software (Wyatt Technology).

## Analytical ultracentrifugation

Sedimentation velocity experiments were conducted using an Optima XL-I analytical ultracentrifuge (Beckman). MICAL1, at concentrations of 1 mg/ml and 5 mg/ml, was prepared in a buffer containing 15 mM Tris-HCl (pH 8.0), 150 mM NaCl, and 2 mM DTT. Centrifugation was performed in double-sector 12 mm centerpieces in an An-60 Ti rotor (Beckman) at 40,000 rpm. Protein sedimentation was monitored through both an absorption optical system and a Rayleigh interference system. The acquired data were subsequently analyzed using SEDFIT software. Additionally, the expected sedimentation coefficients of the structural models were predicted employing WinHydroPRO.

## Biolayer interferometry binding experiments

For biolayer interferometry (BLI) binding experiments, a construct encoding human MICAL1[CC] (residues K918-G1067) was cloned into the pET45b (Merck) vector, incorporating an N-terminal His6 tag and C-terminal Avi tag. MICAL1[CC] was biotinylated in vivo in BL21(DE3)BirA cells, supplemented with 100 μM D-biotin. The biotinylated MICAL1[CC] was produced and purified using the same methods as those used for non-biotinylated MICAL1[CC]. Subsequently, the purified biotinylated MICAL1[CC] was immobilized at a concentration of 8.8 μg/ml onto Octet SAX biosensors (Sartorius).

BLI experiments were conducted using an Octet R8 (Sartorius) in a buffer containing 20 mM HEPES (pH 7.4), 150 mM NaCl, 2 mM DTT, 2 mM MgCl$_2$, and 0.05% (v/v) Tween 20 at 20 °C. The signal from each analyte was corrected by subtracting the signal from the reference biosensor and reference sample. All obtained data were analyzed using the Octet Analysis Studio 12.2.0.20. The BLI experiments were performed in duplicate.

## Circular dichroism spectroscopy

The CD spectra of the CC domain were recorded using a Chirascan-plus spectrophotometer (Applied Photophysics, Leatherhead, UK) with 1 mm quartz cells. Spectral accumulation parameters included a 2 nm bandwidth over the wavelength range of 190–260 nm. Measurements were performed in 1 nm steps, with an averaging time of 1 s per step. The nominal concentration of the CC domain was 2 mg/mL, and the spectra were recorded at room temperature. The buffer was composed of 15 mM Tris-HCl (pH 8.0), 150 mM NaCl, and 2 mM DTT. The CD signal was recorded as ellipticity in millidegrees, with the resulting spectra averaged from two scans and corrected for the buffer spectrum. The ratio of ellipticity at 222 nm to that at 200 nm was averaged and used to assess the secondary structure of the protein.

## F-Actin depolymerization assay

To monitor the depolymerization of rabbit skeletal muscle F-actin in the presence of various MICAL1 proteins, we utilized the Actin Polymerization Biochem Kit™ (Cytoskeleton, Inc). Briefly, pyrene-labeled G-actin was diluted to a concentration of 1 mg/ml by buffer comprising 5 mM Tris-HCl (pH 8.0), 0.2 mM CaCl$_2$, and 0.2 mM ATP. Actin polymerization was initiated by adding a buffer equivalent to 1/40 of the reaction volume, containing 500 mM KCl, 20 mM MgCl$_2$, 50 mM guanidine carbonate, and 10 mM ATP, followed by incubation for 1 hour at room temperature. Subsequently, the F-actin solution was supplemented with MICAL1 proteins, and the reaction was initiated by the addition of 10x concentrated NADPH. The lag time between adding NADPH and starting a measurement was about 20 s. The final concentrations were 200 nM for MICAL proteins and 200 μM for NADPH. The pyrene fluorescence was recorded using Nunc® 96-well black assay plates (Thermo Fisher) on The Spark® Multimode Microplate Reader (Tecan), with excitation and emission wavelengths set at 360 ± 10 nm and 410 ± 5 nm, respectively.

## Single actin filament TIRF microscopy

Microscopy channels were prepared from two HDMS silanised glass coverslips[54] held together by parallel strips of parafilm ~2 mm apart. The coverslips were then heat cured at 60 °C for 15–20 s and gently pressed together. The volume of each channel was approximately 10 μl. To immobilize biotinylated actin filaments to the glass surface, the flow channels were filled with 2 channel volumes (CV) of 20 μg/ml

anti-biotin antibody in 1x PBS (#B3640, Merck) and incubated for 5 min at room temperature. Unbound antibody was washed away by 2 CV of 1% Pluronic-F127 (P2443, Merck) in PBS and left to incubate for 1 hour at room temperature. Before imaging, experimental chambers were washed and equilibrated with 2CV of TIRF imaging buffer (5 mM Tris-HCl (pH 7.8), 0.1 mM CaCl$_2$, 1 mM MgCl$_2$, 50 mM KCl, 0.2 mM EGTA, 1 mM DTT, 0.2 mM ATP, 10 mM DABCO).

Unlabeled G-actin (#ALK99, Cytoskeleton, Inc.), Biotinylated G-Actin (#AB07-A, Cytoskeleton, Inc.) and Rhodamine labeled G-actin (#AR05-B, Cytoskeleton, Inc.) were resuspended according to manufacturer's instruction, mixed in 80:2:20 ratio (Unlabeled:Biotinylated:Rhodamine, at actin concentration 10 mg/mL (~232 μM)), aliquoted to experiment sized volumes, snap frozen in liquid nitrogen, and stored at −80 °C. G-actin was then prepared from frozen aliquots by depolymerization overnight in G-buffer (5 mM Tris-HCl (pH 8.0), 0.2 mM CaCl$_2$, 0.2 mM ATP, 1 mM DTT) at 0.4 mg/ml (~9.3 μM) before spinning at 200,000 × $g$ for 1 h at 4 °C using Beckman Coulter Optima XPN-90 ultracentrifuge (Beckman). Supernatant containing G-actin monomers was taken and protein content was measured using Nano-Drop to estimate G-actin concentration (usually ~7.5 μM). F-actin polymerization was initiated by 10x dilution of the G-actin stock into Actin polymerization buffer (5 mM Tris-HCl (pH 7.8), 0.1 mM CaCl$_2$, 1 mM MgCl$_2$, 50 mM KCl, 0.2 mM EGTA, 1 mM DTT, 0.2 mM ATP) and incubated at 30 °C for 10 min. The F-actin was then diluted to 75 nM with TIRF imaging buffer and immobilized for 4 min in the experiment chambers. Unbound F-actin was then removed by 2CV wash by TIRF imaging buffer. Meanwhile, the MICAL proteins were diluted into TIRF imaging buffer and equilibrated at 30 °C for 5 min. After equilibration, MICAL proteins were supplemented with 200 μM NADPH and 2CV of the mixture was introduced to the channel with immobilized actin filaments. The final concentration of MICAL proteins was 500 nM. This step acted also as a washing step for unbound or unpolymerized actin. The F-actin depolymerization was recorded in TIRF mode right after the protein addition for 10 min in 30 s intervals. All movies were recorded using the same settings on an inverted widefield microscope Nikon Eclipse Ti (Nikon, Tokyo, Japan) equipped with a motorized XY-stage, a perfect focus system, Nikon Apo TIRF 60× Oil, NA 1.49 objective, and a sCMOS Hamamatsu ORCA-flash4.0 LT (Hamamatsu Photonics, Japan).

Time-lapse TIRF movies were analyzed with FIJI package of ImageJ2[55,56]. Briefly, the movies were first corrected for fluorescence bleaching[57], then the background was subtracted using the rolling ball radius algorithm (10 pxls) and subjected to JFilament 2D algorithms[58] to identify, track and measure the single filaments in time. The single filaments were chosen based on specific criteria: a minimum length of 4 μm, no contact with neighboring filaments, and uninterrupted visibility throughout the entire experiment. Depolymerization rate was calculated as a number of actin units lost per second and visualized in Prism 10.0.2 (GraphPad).

### Hydrogen – Deuterium exchange coupled with mass spectrometry (HDX-MS)

The HDX experiment was conducted on the MO domain alone and the MO domain in complex with F-actin. A total of 50 pmol of MO at a concentration of 12.4 μM was used per reaction. The MO-F-actin complex was prepared by mixing MO with F-actin at a 1:2 stoichiometry and incubating the mixture on ice for 45 minutes. These protein samples were then diluted 14-fold with a D$_2$O-based buffer (20 mM HEPES-NaOH, pD 7.5/pH 7.1, 50 mM NaCl, 2 mM DTT). H/D exchange was sampled at 20 s, 2 minutes, 20 min, and 2 h at 20 °C. After these intervals, the reactions were quenched by mixing with 50 μL of cold quench buffer (1 M glycine-HCl, pH 2.3, 6 M urea, 2 M thiourea, 400 mM TCEP-HCl). The 20-second and 20-minute time points were replicated.

For the injection of the reaction mixture into the LC system, a PAL DHR autosampler (CTC Analytics AG) controlled by Chronos software (AxelSemrau) was employed. The LC system included a temperature-controlled box and an Agilent Infinity II UPLC (Agilent Technologies) directly connected to an ESI source of timsTOF Pro (Bruker Daltonics). The injected sample was delivered onto a protease column packed with immobilized AnPep/pepsin in tandem with nepenthesin-2 matrix (bed volume 69 μL) (Affipro) and subsequently onto the trap column (SecurityGuard™ ULTRA Cartridge UHPLC Fully Porous Polar C18, 2.1 mm ID, Phenomenex) under a flow of 0.4% formic acid (FA) in water driven by the 1260 Infinity II Quaternary pump at a flow rate of 200 μL/min. After 3 minutes, desalted peptides were eluted and separated using an analytical column (Luna Omega Polar C18, 1.6 μm, 100 Å, 1.0 ×100 mm, Phenomenex) under a water-acetonitrile (ACN) gradient (10%–45% in 6 min; solvent A: 0.1% FA in water, solvent B: 0.1% FA, 2% water in ACN). The water-ACN gradient was delivered by the 1290 Infinity II LC pump at a flow rate of 50 μL/min. To minimize the loss of deuterium, the LC system was refrigerated to 0 °C.

Fully deuterated control samples used for deuterium back-exchange correction were prepared manually ($n = 3$) using the same LC setup in the offline regime. Desalted peptides were captured, dried, and re-dissolved in D$_2$O-based buffer (pD 7.5), enriched with 1 mM TCEP, and incubated overnight at 37 °C. Samples were then analyzed by the same setup as the partially deuterated samples. The mass spectrometer operated in MS mode with a 1 Hz data acquisition rate. Acquired LC-MS data were peak-picked and exported in DataAnalysis (v. 5.3, Bruker Daltonics) and further processed by the DeutEx software[59]. Data visualization was performed using MSTools[60] and PyMOL 2.5.5. For peptide identification, the same LC-MS system was used, but the mass spectrometer operated in data-dependent MS/MS mode using PASEF. The LC-MS/MS data were searched using MASCOT (v. 2.7, Matrix Science) against a customized database combining sequences of MO, common cRAP.fasta (https://www.thegpm.org/crap/), and the proteases. Search parameters were set as follows: no-enzyme, no modifications allowed, precursor tolerance 10 ppm, fragment ion tolerance 0.05 Da, decoy search enabled, FDR < 1%, IonScore > 20, and peptide length > 5.

### Molecular dynamics simulations

For molecular dynamics simulations, we utilized GROMACS[61] with the CHARMM36m[62] force field. Missing loops were modeled by ColabFold[63], a modified version of AlphaFold[64]. All inputs for the simulation were prepared using the CHARMM-GUI[65] web server, and the FAD ligand was parametrized using the CHARMM General Force Field[66]. For Zn$^{2+}$ binding, we used a non-bonded approach with the metal-coordinating residues in appropriate protonation states (Cym, Hie). The protein was solvated in a box of TIP3P water, with a minimum distance of 1.0 nm from the box edge. Charges were neutralized by the addition of 150 mM NaCl (Supplementary Table 4). Long-range electrostatics were treated with the particle-mesh Ewald summation[67], and H-bond lengths were constrained using the LINCS algorithm. First, the system was energy minimized until the maximal force was lower than 1000 kJ·mol$^{-1}$·nm$^{-1}$. Subsequently, the system was initialized with random velocities and equilibrated in a short NVT simulation with a 1 fs step for 125 ps. Protein and ligand were position-restrained during the minimization and equilibration. For the production simulation, the Nose-Hoover thermostat and the Parrinello−Rahman barostat were used to maintain a temperature of 303.15 K and a pressure of 1 atm. Simulations were carried out in triplicate, with each run lasting 100 ns, for a total simulation time of 300 ns. After evaluating the RMSD against the experimentally determined structure, the first 10 ns of each simulation were excluded. Snapshots were extracted every 1 ns for analysis of contacts and occupancy. For molecular dynamics simulation, the initial coordinates, simulation input files and the final

coordinate output file generated in this study have been deposited in the Zenodo public repository under accession code 13987364.

### Reporting summary

Further information on research design is available in the Nature Portfolio Reporting Summary linked to this article.

## Data availability

The cryo-EM map has been deposited in the Electron Microscopy Data Bank (EMDB) under accession code EMD-50026 and the corresponding model coordinates have been deposited in the Protein Data Bank (PDB) under accession number 9EWY. For molecular dynamics simulation, the initial coordinates, simulation input files and the final coordinate output file generated in this study have been deposited in the Zenodo public repository under accession code 13987364. The HDX data have been deposited to the ProteomeXchange repository with the dataset identifier PXD057312. Source data are provided with this paper.

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

## Acknowledgements

We thank Petr Pompach and Pavla Vankova for their assistance with the HDX-MS experiments and Martin Lepsik for his support with the MD simulations. We also extend our thanks to the staff of the crystallization and imaging facility at Biocev and the cryo-EM facility at Ceitec for their support and for providing access to the necessary instruments for this study. We also acknowledge the funding support from Charles University through the Primus program (Grant No. PRIMUS/21/SCI/003 to D.R.) and the Czech Science Foundation via the Junior Star Grant (Grant No. 21-27204 M to D.R.). We are thankful for the computational resources provided by the e-INFRA CZ project (Grant No. 90254 to D.R.). Z.L. acknowledges the support from the project National Institute for Neurological Research (Programme EXCELES, ID LX22NPO5107 to Z.L.), funded by the European Union - Next Generation EU. N.K. and Karolina K. acknowledge funding from the Charles University project GAUK (338222 to N.K.) and (252859 to Karolina K.) respectively. Further support came from the CF Cryo-EM and CF Crystallization facilities of the CIISB, Instruct-CZ Centre, under the auspices of the Ministry of Education, Youth and Sports of the Czech Republic (Grant No. LM2023042) and the European Regional Development Fund-Project UP CIISB (Grant No. CZ.02.1.01/0.0/0.0/18_046/0015974) and Imaging Methods Core Facility at BIOCEV supported by the Ministry of Education, Youth and Sports of the Czech Republic (Grant No. LM2023050.).

## Author contributions

Conceptualization, DR, MH, and ZL; Methodology, DR, MH, ZL, VV, AS and JN; Investigation, MH, Karolina K, JS, AS, JV, FN, MS, DP, Krystof K, and NK; Formal Analysis, MH, Karolina K, JS, AS, and DP; Writing, DR, Karolina K, and MH, Funding acquisition, DR, and ZL; Supervision, DR, JN, and ZL.

## Competing interests

The authors declare no competing interests.
