## [Transparent Peer Review file · Nature Communications]

Structural basis of MICAL autoinhibition

Corresponding Author: Dr Daniel Rozbesky

Version 0:

Reviewer comments:

Reviewer #1

(Remarks to the Author)

This study describes a cryo-EM structure of MICAL, an enzyme that promotes actin filament disassembly through oxidation of residues Met44 and Met47 near a crucial inter-subunit contact in F-actin. Despite the importance of this MICAL's activity, the structural mechanisms underlying its autoinhibition and activation have remained elusive. The structural data presented here and biochemical analyses (Co-sedimentation, TIRF, MALS assays and MD simulations) contributes to making this an interesting, timely, and rather complete study. Supported by this evidence, the authors propose a mechanism of autoinhibition and activation. I enthusiastically support publication of this study in NC, after addressing some minor concerns.

1. In scientific literature, maintaining a professional tone is essential. The statement, "Overall, the shape of the MICAL1 resembles a classic video game protagonist, Pacman (MO domain), grasping a French baguette (CC domain) topped with tomato (CH domain) and cucumber (LIM domain)," is not appropriate for a reputable scientific journal. While such analogies might be acceptable in informal settings, they are out of place in a formal scientific publication.
2. Similarly, the model depicted in Figure 6 appears to be more akin to a representation from a video game rather than a scientific visualization.
3. I am fully satisfied with the quality of the map. I suggest the authors include a full pic of the map as part of Figure 1, as well as a resolution-colored map in Supplementary materials.
4. The description of cryo-EM data processing and map validation in the Methods section is insufficient.
5. Additionally, crucial validation information and a figure outlining the data processing strategy are missing. For instance, any paper reporting a cryo-EM structure should include the FSC curve and orientation distribution for the map.
6. Please, add numbers to specify domain boundaries in Figure 1a and ensure that domains are drawn to scale.
7. Finally, the insights gained from the MD simulations are not clearly articulated. Many of the results seem obvious and predictable. A better effort should be made to integrate this aspect of the work and its rationale with the overall narrative.

Reviewer #2

(Remarks to the Author)

The article by Horvath and colleagues is an interesting and important one that will be of broad interest. In particular, MICAL family proteins have emerged as important F-actin disassemblers that are expressed broadly and are involved in a number of physiological and pathological processes. Furthermore, MICALs are unusual F-actin disassemblers -- in that they are enzymes that use a Redox mechanism to exert their effects: oxidizing actin on specific residues to induce F-actin disassembly. Notably, previous biochemical, genetic, and cellular work has shown that MICALs are kept in an inactive conformation -- but following binding to specific proteins they become enzymatically active. Yet, the field has a poor physical/structural understanding of this inactive conformation and its activation mechanism. The manuscript by Horvath and colleagues helps provide a physical/structural understanding of the inactive conformation of MICAL and the mechanisms underlying this property of the MICALs. However, several issues need to be addressed before the manuscript is acceptable for publication.

1) Related to the mechanism of autoinhibition: The observations with the CC and the Delta CC in Fig. 3 and 5 needs further support/better presentation.

1A) The results in Fig. 3 are based on two assays: TIRFM (Fig. 3a) and pyrene-actin (Fig. 3b-c, e). A summary of the TIRFM data (Fig. 3a) suggests CC robustly inhibits Delta CC (back into the vicinity of the inhibition seen with the full-length

MICAL1). However, in looking at the specific examples shown in Fig. S5, the Delta CC + CC (100uM) does not show the same inhibition of F-actin disassembly as MICAL1. This should be addressed. Likewise, the pyrene-actin data does not show a similar robust inhibition as that presented in Fig. 3a (e.g., compare the red line in Fig. 3e to the green line in Fig. 3b). Thus, unlike what the authors say “(b, c, e) Data from pyrene-labeled actin depolymerization assays corroborate the findings from single actin filament TIRF microscopy presented in (a).” this does not appear to be the case. This should be addressed including that upon closer examination it can be seen that 25uM and 100uM concentrations of CC were used in with the TIRF experiments (Fig. 3a), but only [10uM] at the highest was used in the pyrene actin assays (Fig. 3e). It is not clear why the authors performed these two different experimental approaches with such widely different concentrations of CC. Is this the reason for the modest ability of the CC to inhibit the Delta CC in the pyrene actin assays or is it a difference in the assays/analyses? Thus, experiments with [25uM] and [100uM] of CC with Delta CC should be done for the pyrene actin assays (Fig. 3e, and also Fig. 3c with MO). The authors also show in the TIRFM experiments in Fig. 3a that the addition of the CC alone (even at these higher concentrations) did not affect the depolymerization rate (and they also show this because of CC’s lack of effect on the MO in these assays). Likewise, there is data for no effect of the CC alone in the pyrene actin assays (Fig. 3b) but the concentration is not indicated. Thus, experiments should also be done for the higher concentrations of CC alone in the pyrene actin assays. Further, these results with CC alone should also be presented in with Figs 3c and e (and not Fig. 3b), so it is readily apparent for comparison with the data presented in each of those graphs. Also, the concentrations of MICAL1, MO, Delta CC, NADPH, and actin should be listed in the legend of Fig. 3 so that it is seen that they have been added at similar concentrations (e.g., in the pyrene actin methods it says 500nM was added for MICALs and 200uM for NADPH but I did not see the [actin], I did not see the concentrations used in the TIRF experiments).

1B) Related issues are present in Fig. 5a-c. For example, how much CC is added in Fig. 5a-c? And if CC were added at higher concentrations (25-100uM) would this inhibit Delta CC’s sedimentation with F-actin to the same extent as MICAL1 (Fig. 5c). In short, for the reader to better interpret the data in Figs. 3 and 5, the authors need some consistency among the different assays (Figs. 3a, 3c, 3e, and 5a-c) related to the CC. The protein concentrations should also be listed in the legend.

1C) Depending on their results, the authors may need to address the reasons why the CC might not inhibit the Delta CC back to the same level as MICAL1 (full-length MICAL1). See also Comment #2A, below.

1D) Note that the authors say “The addition of the purified CC domain led to a depolymerization rate inhibition by 41.4% and 52.5% at CC concentrations of 25 and 100 μM, respectively (Fig. 3a, e)”. This is clearly not the case for what is presented in Fig. 3e.

1E) The authors say in relation to Fig. 5a-c: “The addition of the purified CC domain did not substantially alter binding of the MO domain to F-actin relative to the loading control” and this is consistent with the data in Fig. 3a, c. However, in the graph in Fig. 5c the authors report it as a significant difference (please note the asterisk). Is the asterisk a typo in the graph? If not, what is the reason behind the different effects in Figs. 3a, c and 5a.

2) Additional issues related to the mechanism of autoinhibition:

2A) The authors results bring up the question of why CC needs to be added at such high concentrations relative to the Delta CC to get an effect? The binding data in Fig. 3d ($KD = 0.58 \pm 0.17 \mu M$) also argues against the need of these super high concentrations. This should be discussed in the text.

2B) On my first reading, I was struck with how much the authors say that MICAL1 pellets with F-actin (i.e., full-length autoinhibited MICAL1 pellets with F-actin). In particular, the authors say in the results “Although the purified CC domains did not exhibit direct binding to F-actin, approximately 21.8 ± 1.7% of the full-length MICAL1 was pelleted with actin (Fig. 5 a-c).” My first thought was that this is >20% of the autoinhibited MICAL and thus does not match to their contention that it has little to no activity (Fig. 3a-b and throughout the manuscript). However, after a closer look at comparing their data in Fig. 5a (with F-actin) and Fig. 5b (without F-actin), it is clear that there is pelleting without F-actin. There may also be a little more loaded/darker staining in Fig. 5a. In any case, there is MICAL1 in the pellet without F-actin (Fig. 5b) and so this should be factored into their statement “approximately 21.8 ± 1.7% of the full-length MICAL1 was pelleted with actin”. Therefore, the labeling of the graph in Fig. 5c needs to be updated – since it is not the same as “pelleted with F-actin”, as said in the results. In short, if the authors want to say something in the text about the pelleting with F-actin they need to subtract out the pelleting on its own and present that in a graph. Moreover, upon comparing the amount of the pellet vs soluble in Fig. 5a, it looks like much less than 20% of MICAL1 is in the pellet compared to the amount in the soluble (and that would be even less when the amount in the pellet in 5b is subtracted). For example, the difference between S and P for the Delta CC (Fig. 5a) seems to match somewhat to what is presented the graph (Fig. 5c), but the difference between the S and P for MICAL1 does not match.

2C) The manner in which the authors discuss the CC binding and inhibiting the MO does not match the data. In particular, the authors data support (and they state) that the CC does not inhibit the MO (Fig. 3a, 3c, 5a-c) and the CC does not bind to the MO (Fig. 3d). Yet, within the text they contradict their results. For example, the title of Fig. 5 is “F-actin and the CCα1 helix compete for binding to the MO domain”. Likewise, in the legend of Fig 6 they say “Initially, MICAL1 activity is autoinhibited by an intramolecular interaction between the MO domain and the CCα1 helix. This interaction sterically hinders F-actin from binding to the MO domain.” So too, in the results they say “Thus, it appears that the binding site of the CCα1 helix on the MO domain either fully or partially overlaps with the F-actin binding site.” And in the discussion they say “Our co-sedimentation experiments suggest that the binding site of the CCα1 helix on the MO domain either fully or partially overlaps with the F-actin binding site.” Thus, these statements do not match their data and so they need to be revised. Further, the authors results show that the MO is sufficient to bind to F-actin (Fig. 5a) but the MO is not sufficient to bind to the CC (Fig. 3d). Thus,

the authors statement that “the binding site of the CC α 1 helix on the MO domain either fully or partially overlaps with the F-actin binding site” is not the accurate. The authors need to rework these statements and any others that give this mixed message. See also Comment #2D.

2D) Elsewhere the authors say “These findings suggest the CC domain alone cannot autoinhibit the MO domain's depolymerization activity, indicating that autoinhibition requires additional elements from the L1-CH-L2-LIM-L3 region.” This statement by the authors is supported by their structure of the full-length MICAL1 in which (as they say in the results) they see interactions between the CH, LIM, and L2 α 1 linker regions and the CC. Indeed, the authors say “These data suggest that the CC domain, not alone but particularly in association with the tripartite CH-L2 α 1-LIM assembly, exhibits a competitive binding interaction with F-actin for the MO domain.”, “In summary, the tripartite assembly of CH-L2 α 1-LIM plays a pivotal role in the binding of the CC domain to the MO domain, enabling the CC domain to adopt a conformation that binds the MO domain and inhibits its catalytic function.”, and “Our findings, supported by biochemical and functional analyses, indicate that MICAL1 autoinhibition hinges on the binding of the CH and LIM domains, facilitated by a helical region of the long linker, to the CC domain. This interaction maintains the CC domain in a specific conformation, enabling it to interact with the catalytic domain and thus enforce autoinhibition.” In light of these statements (and many others that are similar) by the authors, it is important to test for this type of mechanism. In particular, the authors should generate the protein they describe, i.e., containing the CH-LIM-CC domains (i.e., all the regions of MICAL1 that the authors mention that are C-terminal to the MO) and test it for its ability to inhibit the MO. This can be done relatively easily using one of the assays that they authors have already employed here (TIRF microscopy, pyrene-actin, or F-actin sedimentation assays). This will provide a specific experiment to further test their hypothesis of the mechanism of MICAL1's autoinhibition.

3) Related to the mechanism of “activation”. It is clear that the authors have uncovered structural mechanisms related to the autoinhibition of MICAL proteins, but there is little structural information on the activation mechanisms. The authors have formulated their activation model based on modeling of previous structures of fragments of MICAL1 versus having a structure of the full-length autoinhibited MICAL and structure of the full-length MICAL in the presence of Rab to compare or other similar types of structural experiments and experimental testing of those ideas/observations. For example, the authors say based on their modeling “the CC high-affinity binding site for Rab10 in the complex overlaps with the binding site for the tripartite assembly CH-L2 α 1-LIM (Fig. 4c), indicating mutually exclusive binding to the CC domain.” and “the axial tilt of CC α 3 in autoinhibited MICAL1 precludes low-affinity Rab10 binding (Fig. 4c). Yet, it is not clear how the binding of Rab would displace/override/dissociate the binding of the tripartite assembly and the autoinhibitory interaction with the MO. Indeed, the authors also state their uncertainty regarding other areas related to activation including: “While it remains uncertain whether this results in a complete or partial dissociation of the CC domain from the MO domain, this conformational shift likely exposes the F-actin binding site. Additionally, the conformation of the Rab-bound CC domain may be further stabilized by a second Rab molecule, enhancing the activation process.” Thus, while their modeling provides interesting insights into the mechanism of activation and I don't believe that further experimentation in this direction is needed for this paper, the authors should scale back their use of activation in the title, abstract, and paper. Perhaps a title of “Structural basis of MICAL autoinhibition with insights into its activation” or something similar.

4) The authors make multiple statements in the text but don't cite a figure or present a figure to support their statement. It would help the general audience for the authors to include this. For example,

4A) “In the cryoEM structure of MICAL1, the CH domain primarily interacts with the proximal part CC α 2 helix through a combination of hydrophilic and hydrophobic interactions.”  The authors need to refer to a figure/part of a figure or present a more detailed figure to support their statement.

4B) “Consistent with earlier predictions, the LIM domain (residues 695-757) features two contiguous zinc fingers, separated by a Phe-Arg pair. The LIM domain engages the central part of the CC α 1 and CC α 2 mainly through hydrophobic contacts. It also binds to the CH domain, with hydrophobic interactions between Phe716 from LIM and Val585 and Val586 from CH, further cemented by a salt bridge (Asp596 - Arg709). Intriguingly, the LIM domain also forms van der Waals contacts with the region between MO β 11 and MO α 11, crucial for the stability of the MO-CC domain interaction.”  The authors need to refer to a figure/part of a figure or present a more detailed figure to support their different statements here.

4C) “In the cryoEM structure, the CH domain binds to the CC domain.”  The authors need to refer to a figure/part of a figure or present a more detailed figure to support their statement.

5) Some additional labeling changes/updates should be made to the figures and text:

5A) In Fig. 1b  label each of the CC helices

5B) The authors say in the results “Surprisingly, the addition of the purified CC domain at varying concentrations failed to inhibit depolymerization (Fig. 3a, c)”  but looking at the context it should be modified to “.....the addition of the purified CC domain at varying concentrations to the MO failed to inhibit depolymerization.....”

5C) For Fig. 5c: it should indicate on the figure that is with F-actin. For Fig. 5A and 5B, it would also be good to indicate on the figure that it was with F-actin and without F-actin, respectively

6) Statements/sentences in the manuscript need to cite references (so that the reader is not left wondering what is the basis of these statements). For example,

6A) "Similar to the previously published CC structures,"  missing the citations (refs 26 and 27)

6B) "This domain is highly similar to that of the CH domain of human MICAL1 determined via NMR."  missing the citation (Solution structure of calponin homology domain of Human MICAL-1. Hongbin Sun, Haiming Dai, Jiahai Zhang, Xianju Jin, Shangmin Xiong, Jian Xu, Jihui Wu & Yunyu Shi; Journal of Biomolecular NMR (2006) 36:295–300)

Reviewer #3

(Remarks to the Author)

The present manuscript delves into an investigation of the molecular mechanisms underlying the regulation of MICAL1, a protein crucial for cellular dynamics through its role in actin filament disassembly. By integrating cryo-EM structural analysis, alongside biochemical and functional assays, the authors provide comprehensive insights into MICAL1's autoinhibition and activation mechanisms.

The study reveals the cryo-EM structure of human MICAL1 at a nominal resolution of 3.1 Å, highlighting that autoinhibition is mediated by an intramolecular interaction between its N-terminal catalytic and C-terminal coiled-coil domains, which obstructs F-actin interaction. Moreover, the authors elucidate that allosteric changes in the coiled-coil domain, coupled with the binding of the CH-L2 α 1-LIM domains to this coiled-coil region, are critical for MICAL1 activation and autoinhibition. Furthermore, the findings suggest that these regulatory mechanisms are evolutionarily conserved, indicating a potential universality across the MICAL protein family. This study significantly advances the understanding of MICAL1 regulation and provides a foundational basis for future research into its role in cellular dynamics and potential implications in disease contexts where actin dynamics are disrupted.

Comments:

1. The authors utilized AlphaFold to model the linker regions and conducted a 100 ns molecular dynamics simulation. However, a more in-depth discussion on the reliability and accuracy of these models is necessary. Have additional validations been performed, such as comparisons with experimental data or verification through other computational methods? These validations would help strengthen the interpretation of the models and the credibility of the results.
2. The authors used AlphaFold for modeling the linker regions and conducted a 100 ns molecular dynamics simulation, describing the interactions between the four domains and the linker regions. It is recommended that the authors include illustrations to depict and describe the movement patterns and correlations of each domain and the linker regions.
3. In Figure S3(b), RMSF of C α atoms (orange) with standard deviation depicted in grey. I believe this should be RMSD. Please verify.
4. The authors mention that the linker regions exhibit significant flexibility and explore a wide conformational space, which plays a crucial role in the function and structure of MICAL1. It would be beneficial if the authors could provide a detailed description of the different conformations of the linker regions observed during the simulation and identify which conformations are particularly relevant to biological functions. Additionally, the authors noted that certain segments exhibit relative rigidity, suggesting an association with the core structure. Please explain the reasoning behind this association with the core.
5. The simulations describe the MO, CH, LIM, and CC domains as relatively rigid entities, with some highly flexible local regions, particularly in the loop region connecting the helices of the CC domain. Please explain why the loop region connecting the helices of the CC domain exhibits higher flexibility.
6. The authors state that the binding of Rab10 to MICAL1 results in the dissociation of the CH-L2 α 1-LIM complex. Please elaborate on how this dissociation process is regulated and clarify whether it is a direct or indirect result.
7. Regarding how the conformation of the Rab-bound CC domain further stabilizes the activation process, could the authors provide more details or a theoretical explanation?

Version 1:

Reviewer comments:

Reviewer #1

(Remarks to the Author)

The authors have satisfactorily addressed the criticism. I believe the revised MS is acceptable for publication NC

Reviewer #2

(Remarks to the Author)

The authors have done a nice job of revising the manuscript, including conducting critical functional assays (Fig. 3f) and providing clearer insights into the questions addressed in Fig. 5. However, the authors inadequately addressed a few of my initial comments and thus still need to address a couple of their findings in the paper

1) The major issue that I mentioned in the first version of this manuscript is the relatively poor job that the CC alone does in inhibiting the Delta CC construct. However, instead of adequately responding to my comments and addressing this in the manuscript, the authors focused on addressing the differences between the TIRFM and pyrene assays, which were already understood and not the point I was getting at. Therefore, the manuscript still needs work in this area to accurately reflect the authors data.

1A) The relatively weak inhibition seen in the TIRFM images in Fig. S6 (e.g., compare MICAL1Delta CC + CC (100uM) with

MICAL1Delta CC and also to MICAL1, which is ~fully inhibited) roughly matches the weak inhibition of the CC on the Delta CC (Fig. 3e, compare light red to dark red and also to black, which is ~fully inhibited). Thus, given that these are different assays, I think the relative inhibition in images of the TIRFM data (Fig. S6) is comparable to the relative inhibition of the pyrene actin assays (Fig. 3e). However, instead of the authors seeing the point of my questions (i.e., that the CC is a weak and not a robust inhibitor of the Delta CC), they unnecessarily discussed the differences between the TIRFM and Pyrene actin assays as well as other methodological reasons why this is the case. Of course, these two assays are different but they have been used complimentary for hundreds/thousands of papers over the years. Instead, the authors should follow their data, which say that the CC is not a strong inhibitor of the Delta CC and discuss why that might be the case. Given the other robust data that the authors have now provided (Fig. 3f), it seems likely that once the autoinhibition of MICAL1 is relieved, the CC is not robustly/fully able to inhibit MICAL1's activity again? Thus, as I specifically detail below, this section of the manuscript needs to be reworked to focus on the point that the authors experiments reveal vs simply pointing out differences in methodologies.

1B) Lines 196-197: Notably, the addition of the purified CC domain to MICAL1ΔCC led to a marked inhibition of F-actin depolymerization (Fig. 3a, e).  I would recommend either removing the term "marked" or changing it to "noticeable". Having done a number of TIRFM and pyrene assays, I would not say it is marked inhibition but it is convincing. And coupled with Fig. 3f, it convinces the reader that the authors are on the right track. Referring to the actual data in Fig. S6 should also be done in this sentence (Figs. 3a, 3e, S6)

1C) The authors helpfully changed the manuscript to use a ratio in Fig. 3 but did not make the change to ratio in Fig. S6. This should be done in Fig. S6 so it is easier for the reader to go between the two figures.

1D) Lines 197-200: "Data from the pyrene-labeled actin depolymerization assays (Fig. 3e) were consistent with the trends observed in the single actin filament TIRF microscopy (Fig. 3a), though the degree of inhibition observed in the pyrene assays was less pronounced, likely due to inherent methodological differences between the two techniques." The authors should remove "likely due to inherent methodological differences between the two techniques" since they don't know that. Or replace "likely" with "possibly". However, as mentioned in 1A), I don't see much relative difference in inhibition between the two assays so they could alternatively remove ", though the degree of inhibition observed in the pyrene assays was less pronounced, likely due to inherent methodological differences between the two techniques.". The authors also need to refer to Fig. S6 (the actual images) anytime they refer to Fig. 3a in the results.

1E) Lines 200-204: "Interestingly, even at CC domain concentrations near saturation (94.4% bound at 10 μM, as calculated based on the determined KD), the purified CC domain did not inhibit MICAL1ΔCC to the same extent as full-length MICAL1. This discrepancy could be attributed to multiple CC conformations in solution or an overestimation of the KD in the BLI experiment due to the absence of F-actin." I would suggest that there are also other explanations that the authors might want to touch on here or elsewhere (see point 1F).

1F) The data in Fig. 3f are robust; thereby helping to validate the authors proposed model of autoinhibition. Thus, it would seem that there are likely other reasons why the CC does not robustly/fully inhibit the Delta CC vs what the authors mention in Lines 200-204. The authors should at least discuss this as I allude to in 1A) including perhaps that without being connected to more N-terminal regions of MICAL, the CC alone does not adopt the right conformation to robustly/fully inhibit the MO, and/or once the autoinhibition of MICAL is relieved, the CC is not sufficient to robustly/fully inhibit it again, and/or perhaps other reasons. I think this is important to discuss vs simply stating it is due to issues with overestimation of KD, CC solution conformations, or F-actin assays, which is clearly not the case for what they are observing in the pyrene actin assays with MO + delta MO compared to Delta CC + CC.

1G) Lines 731-733: "(b, c, e) Data from pyrene-labeled actin depolymerization assays demonstrate overall trends that are consistent with the findings from TIRF microscopy (a). However, the degree of inhibition observed is less pronounced, likely due to inherent differences between the two methods." This is not an accurate statement: the MO depolymerization is very pronounced using both assays and the inhibition of the MO by the Delta MO is very pronounced in the pyrene actin assays (much more so than the CC inhibition of the Delta CC). So too, same points as above, TIRFM and pyrene actin assays have been used as complimentary approaches for years, so it is not necessary to mention inherent differences between the methods. Therefore, I would remove the statement "However, the degree of inhibition observed is less pronounced, likely due to inherent differences between the two methods."

Minor:

2) Line 104: "A key feature of the MO-CC interaction involves a region between MOβ17 and MOα15..."  should be modified to include Fig. S1. So that the reader knows where to access these data: "A key feature of the MO-CC interaction involves a region between MOβ17 and MOα15 (Figs. 1d, S1)..."

3) A reference to Fig. S1 should also be added to Lines 107-110 so that it reads "...and Ile931 from the CCα1 and CCα2 helices (Figs. 1e, S1)"

4) Lines 135-138: "The shortest linker, L1 135 (residues 490-507), serves as a bridge between the MO and CH domains. The intermediate L2 linker 136 (residues 613-694) connects the CH domain to the LIM domain, and the longest, L3 (residues 758 to 137 907), links the LIM to the CC domain"  Refer to Fig. 1a at the end of this sentence

5) In the legend of Fig. S4c the authors use cyan to indicate a region in the MO. However, they also use Cyan to indicate the

L1 region in Fig. 2a. So, I would recommend changing the color of the cyan in Fig. S4c (so it is not misunderstood that they are referring to a region of the L1, which I originally thought perhaps they were).

6) For ease to help the reader, please reorder Fig. 3a so that it matches the description of the constructs in the legend from top to bottom (not bottom to top)

7) Please note a few typos such as: Line 813 "(c) Visualization of the of MO domain with CC α 1 Helix."  change to "Visualization of the MO domain with CC α 1 Helix."

Reviewer #3

(Remarks to the Author)

I think the manuscript is now ready for the publish on Nature Communications.

Version 2:

Reviewer comments:

Reviewer #2

(Remarks to the Author)

The authors have carefully addressed my remaining concerns. I believe the paper is ready for publication.

Title: Structural basis of MICAL autoinhibition

Tracking number: NCOMMS-24-22666

Authors: Matej Horvath, Adam Schrofel, Karolina Kowalska, Jan Sabo, Jonas Vlasak, Farahdokht Nourisanami, Margarita Sobol, Daniel Pinkas, Krystof Knapp, Nicola Koupilova, Jiri Novacek, Vaclav Veverka, Zdenek Lansky and Daniel Rozbesky

August 15, 2024

Dear Reviewers,

Thank you very much for your interest in our work and for your insightful and constructive comments. We have carefully considered all your suggestions, and we believe that your feedback has improved the quality and clarity of our manuscript.

We appreciate the time and effort you have dedicated to reviewing our paper. Please find below our point-by-point responses to your comments.

REVIEWER COMMENTS

Reviewer #1 (Remarks to the Author):

This study describes a cryo-EM structure of MICAL, an enzyme that promotes actin filament disassembly through oxidation of residues Met44 and Met47 near a crucial inter-subunit contact in F-actin. Despite the importance of this MICAL's activity, the structural mechanisms underlying its autoinhibition and activation have remained elusive. The structural data presented here and biochemical analyses (Co-sedimentation, TIRF, MALS assays and MD simulations) contributes to making this an interesting, timely, and rather complete study. Supported by this evidence, the authors propose a mechanism of autoinhibition and activation. I enthusiastically support publication of this study in NC, after addressing some minor concerns.

1. In scientific literature, maintaining a professional tone is essential. The statement, "Overall, the shape of the MICAL1 resembles a classic video game protagonist, Pacman (MO domain), grasping a French baguette (CC domain) topped with tomato (CH domain) and cucumber (LIM domain)," is not appropriate for a reputable scientific journal. While such analogies might be acceptable in informal settings, they are out of place in a formal scientific publication.

Response: Thank you for your valuable feedback. We agree that the analogy used in the original manuscript was informal and not suitable for a formal scientific publication. As a result, we have completely removed this statement to ensure the professional tone of the manuscript is maintained.

2. Similarly, the model depicted in Figure 6 appears to be more akin to a representation from a video game rather than a scientific visualization.

Response: We have revised Figure 6 to ensure it aligns with the standards of scientific visualization. The updated figure now presents the model in a more professional and scientifically appropriate manner.

3. I am fully satisfied with the quality of the map. I suggest the authors include a full pic of the map as part of Figure 1, as well as a resolution-colored map in Supplementary materials.

Response: Thank you for your positive feedback and suggestions. We appreciate your satisfaction with the quality of the map. In response to your recommendation, we have included a full picture of the map as part of Figure 1 and added a resolution-colored map in the Supplementary materials, please see Fig. S2.

4. The description of cryo-EM data processing and map validation in the Methods section is insufficient.

Response: Thank you for pointing out the insufficiency in the description of cryoEM data processing and map validation in the Methods section. We have expanded this section to provide a more detailed and comprehensive description of the procedures and validation techniques used. This includes specifics on data acquisition, processing parameters, software tools, and validation metrics to ensure clarity and reproducibility of our results.

5. Additionally, crucial validation information and a figure outlining the data processing strategy are missing. For instance, any paper reporting a cryo-EM structure should include the FSC curve and orientation distribution for the map.

Response: We have addressed this by including crucial validation information and a new figure outlining the data processing strategy in the revised manuscript. Specifically, we have added the FSC curve and orientation distribution for the map to provide a comprehensive overview of the data validation (please see Fig. S2).

6. Please, add numbers to specify domain boundaries in Figure 1a and ensure that domains are drawn to scale.

Response: Thank you for your helpful suggestion. We have updated Figure 1a to include numbers specifying the domain boundaries and ensured that the domains are drawn to scale.

7. Finally, the insights gained from the MD simulations are not clearly articulated. Many of the results seem obvious and predictable. A better effort should be made to integrate this aspect of the work and its rationale with the overall narrative.

Response: Thank you for your feedback. We acknowledge that the insights from the MD simulations were not clearly articulated in the initial manuscript. To address this, we have revised the relevant sections to better integrate the rationale and findings of the MD simulations with the overall narrative of the study. Additionally, we have taken into account the specific questions and comments from Reviewer 3 regarding the MD simulations to further refine and clarify this aspect of our work. Please see our responses to the Reviewer 3.

Reviewer #2 (Remarks to the Author):

The article by Horvath and colleagues is an interesting and important one that will be of broad interest. In particular, MICAL family proteins have emerged as important F-actin disassemblers that are expressed broadly and are involved in a number of physiological and pathological processes. Furthermore, MICALs are unusual F-actin disassemblers -- in that they are enzymes that use a Redox mechanism to exert their effects: oxidizing actin on specific residues to induce F-actin disassembly. Notably, previous biochemical, genetic, and cellular work has shown that MICALs are kept in an inactive conformation – but following binding to specific proteins they become enzymatically active. Yet, the field has a poor physical/structural understanding of this inactive conformation and its activation mechanism. The manuscript by Horvath and colleagues helps provide a physical/structural understanding of the inactive conformation of MICAL and the mechanisms underlying this property of the MICALs. However, several issues need to be addressed before the manuscript is acceptable for publication.

1) Related to the mechanism of autoinhibition: The observations with the CC and the Delta CC in Fig. 3 and 5 needs further support/better presentation.

1A) The results in Fig. 3 are based on two assays: TIRFM (Fig. 3a) and pyrene-actin (Fig. 3b-c, e). A summary of the TIRFM data (Fig. 3a) suggests CC robustly inhibits Delta CC (back into the vicinity of the inhibition seen with the full-length MICAL1). However, in looking at the specific examples shown in Fig. S5, the Delta CC + CC (100uM) does not show the same inhibition of F-actin disassembly as MICAL1. This should be addressed. Likewise, the pyrene-actin data does not show a similar robust inhibition as that presented in Fig. 3a (e.g., compare the red line in Fig. 3e to the green line in Fig. 3b). Thus, unlike what the authors say “(b, c, e) Data from pyrene-labeled actin depolymerization assays corroborate the findings from single actin filament TIRF microscopy presented in (a).” this does not appear to be the case. This should be addressed including that upon closer examination it can be seen that 25uM and 100uM concentrations of CC were used in with the TIRF experiments (Fig. 3a), but only [10uM] at the highest was used in the pyrene actin assays (Fig. 3e). It is not clear why the authors performed these two different experimental approaches with such widely different concentrations of CC. Is this the reason for the modest ability of the CC to inhibit the Delta CC in the pyrene actin assays or is it a difference in the assays/analyses? Thus, experiments with [25uM] and [100uM] of CC with Delta CC should be done for the pyrene actin assays (Fig. 3e, and also Fig. 3c with MO).

The authors also show in the TIRFM experiments in Fig. 3a that the addition of the CC alone (even at these higher concentrations) did not affect the depolymerization rate (and they also show this because of CC's lack of effect on the MO in these assays). Likewise, there is data for no effect of the CC alone in the pyrene actin assays (Fig. 3b) but the concentration is not indicated. Thus, experiments should also be done for the higher concentrations of CC alone in the pyrene actin assays. Further, these results with CC alone should also be presented in with Figs 3c and e (and not Fig. 3b), so it readily apparent for comparison with the data presented in each of those graphs. Also, the concentrations of MICAL1, MO, Delta CC, NADPH, and actin should be listed in the legend of Fig. 3 so that it is seen that they have been added at similar concentrations (e.g., in the pyrene actin methods it says 500nM was added for MICALs and 200uM for NADPH but I did not see the [actin], I did not see the concentrations used in the TIRF experiments).

Response: We appreciate the reviewer's insightful observation regarding the apparent discrepancy between the pyrene-actin data and the TIRF microscopy data. We acknowledge that the pyrene-actin depolymerization assay does not exhibit the same degree of robust inhibition as observed in the TIRF microscopy data, specifically when comparing the red line in Fig. 3e with the green line in Fig. 3b. We agree that the statement in our manuscript, "(b, c, e) Data from pyrene-labeled actin depolymerization assays corroborate the findings from single actin filament TIRF microscopy presented in (a)," was too strong and did not fully capture the nuances between the two methods. We think that several factors contribute to the differences observed between the pyrene-actin assay and TIRF microscopy data:

i) Sample Size:

While TIRF microscopy allows for high-resolution observation of individual actin filaments, it inherently involves a limited sample size. In contrast, the pyrene-based assay is a bulk technique that simultaneously analyzes the depolymerization of a large population of actin filaments in solution. As a result, TIRF microscopy provides detailed qualitative and quantitative insights into the behavior of individual filaments, whereas the pyrene assay offers a broader quantitative assessment of filament dynamics across the entire population.

ii) Filament Selection:

The pyrene-labeled actin assay provides an averaged measurement of all filaments in the sample, without discrimination. Conversely, TIRF microscopy involves the deliberate selection of filaments for analysis. In our experiments, filaments were chosen based on specific criteria: a minimum length of 4 μm , no contact with neighboring filaments, and uninterrupted visibility throughout the entire experiment. This selection process, while necessary for accurate tracking, could introduce a bias towards filaments with certain characteristics, potentially explaining some of the differences observed between the two methods.

iii) Experimental Setup:

The experimental setup for each method differs significantly. In the TIRF microscopy experiments, actin filaments were immobilized on a surface, allowing for the detailed observation of their dynamics near the surface. On the other hand, the pyrene-based assay measured filaments freely suspended in solution. As TIRF microscopy primarily focuses on surface-adjacent filaments, it may not fully capture the behavior of filaments in a bulk solution, where factors such as filament-filament interactions and diffusion could influence the overall depolymerization kinetics.

iv) Actin Labeling:

The type of fluorescent label used in each method also differs. The pyrene-based assay utilized pyrene-labeled actin, which shifts fluorescence emission upon polymerization, while the TIRF microscopy experiments employed rhodamine-labeled actin. These different labeling strategies could contribute to variations in the sensitivity and dynamics observed, particularly as they relate to the photophysical properties of the labels and their interaction with the actin filaments. Additionally, pyrene labeling at Cys374 is known to produce actin filaments that are less stable compared to the wild type. According to the manufacturer, pyrene-labeled F-actin is stable for only one hour at room temperature, whereas wild-type F-actin remains stable for several days. This difference in stability could further influence the experimental outcomes.

To address this, we have revised the manuscript to clarify that while the overall trends observed in the pyrene-actin assays are consistent with the findings from TIRF microscopy, the degree of inhibition is less pronounced due to the inherent differences between the two methods. Please see the Results section.

We appreciate the reviewer's careful attention to the differences in CC concentrations used between the TIRF experiments (Fig. 3a) and the pyrene actin assays (Fig. 3e). The variation in concentrations between these experiments was intentional and based on maintaining similar molar ratios between MICAL and CC in both assays.

In the pyrene-labeled actin assays, MICAL was used at a concentration of 200 nM, with CC present in excess at 2 μ M, 5 μ M, and 10 μ M, corresponding to molar ratios of 1:10, 1:25, and 1:50, respectively. For the TIRF microscopy experiments, we initially tested a MICAL concentration of 200 nM, matching that used in the pyrene assays, but observed no substantial effect on actin depolymerization when using MO. To detect a significant effect, we found it necessary to increase the MICAL concentration to 500 nM, a 2.5-fold increase. Consequently, we used this higher concentration for all MICAL constructs in the TIRF experiments. The CC concentrations were then adjusted accordingly, with excess concentrations of 25 μ M and 100 μ M, corresponding to molar ratios of 1:50 and 1:200, respectively. Lower molar ratios (below 1:50) did not result in substantial inhibition in the TIRF experiments, which led to the use of higher CC concentrations. While the trends observed in both methods are consistent, we acknowledge that there are discrepancies between the TIRF experiments and the pyrene-labeled actin assays. As discussed before, we believe these differences reflect the inherent variations in the sensitivity and nature of the two techniques, rather than being solely due to the concentration of CC.

To address these points, we have revised Figure 3 to display molar ratios in all graphs, while the total protein concentrations, and actin concentrations are provided in the figure legend. Additionally, we have expanded the Results section to include a comparative evaluation of the outcomes from the TIRF experiments and the pyrene-based assays, offering further clarity on how the results align and where they differ.

1B) Related issues are present in Fig. 5a-c. For example, how much CC is added in Fig. 5a-c? And if CC were added at higher concentrations (25-100 μ M) would this inhibit Delta CC's sedimentation with F-actin to the same extent as MICAL1 (Fig. 5c). In short, for the reader to better interpret the data in Figs. 3 and 5, the authors need some consistency among the different assays (Figs. 3a, 3c, 3e, and 5a-c) related to the CC. The protein concentrations should also be listed in the legend.

Response: We have now added the protein concentrations to the legend of Fig. 3. In light of the limitations associated with the co-sedimentation assay, we have decided to remove the co-sedimentation results originally presented in Fig. 5. These results have been replaced with more robust and reliable data obtained from Hydrogen-Deuterium Exchange (HDX) experiments. For further details, please see our response to point 2B.

1C) Depending on their results, the authors may need to address the reasons why the CC might not inhibit the Delta CC back to the same level as MICAL1 (full-length MICAL1). See also Comment #2A, below.

Response: Please see our response to comment 2A.

1D) Note that the authors say “The addition of the purified CC domain led to a depolymerization rate inhibition by 41.4% and 52.5% at CC concentrations of 25 and 100 μ M, respectively (Fig. 3a, e)”. This is clearly not the case for what is presented in Fig. 3e.

Response: Thank you for your careful review and for identifying the discrepancy between our statement and the data presented in Fig. 3e. We apologize for this oversight. The inhibition percentages we originally reported were calculated from the TIRF experiment data, and we recognize that these values do not accurately reflect the results from the pyrene-labeled actin assay shown in Fig. 3e. This discrepancy was unintentional and likely stems from the inherent differences between the TIRF experiment and the pyrene-labeled actin assay (please refer to our response to comment 1A for further details). We have revised the manuscript to correct this statement, ensuring that the text now accurately reflects the data presented in Fig. 3e.

1E) The authors say in relation to Fig. 5a-c: “The addition of the purified CC domain did not substantially alter binding of the MO domain to F-actin relative to the loading control” and this is consistent with the data in Fig. 3a, c. However, in the graph in Fig. 5c the authors report it as a significant difference (please note the asterisk). Is the asterisk a typo in the graph? If not, what is the reason behind the different effects in Figs. 3a, c and 5a.

Response: In light of the limitations associated with the co-sedimentation assay, we have decided to remove the co-sedimentation results originally presented in Fig. 5. These results have been replaced with more robust and reliable data obtained from Hydrogen-Deuterium Exchange (HDX) experiments. For further details, please see our response to point 2B.

2) Additional issues related to the mechanism of autoinhibition:

2A) The authors results bring up the question of why CC needs to be added at such high concentrations relative to the Delta CC to get an effect? The binding data in Fig. 3d ($K_D = 0.58 \pm 0.17 \mu$ M) also argues against the need of these super high concentrations. This should be discussed in the text.

Response: Thank you for your comment regarding the need for high concentrations of CC relative to MICAL1^{ACC}, and for raising the question of why CC might not inhibit MICAL1^{ACC} to the same extent as full-length MICAL1.

First, we would like to clarify that we do not think the concentrations used in our experiments were super high. For instance, in the pyrene-labeled actin depolymerization assay, MICAL1^{ACC} was used at a concentration of 200 nM, while the purified CC was used at 2 μ M, 5 μ M, and 10 μ M. Given the determined K_D of 0.58 μ M, and considering the tight binding regime between MICAL1^{ACC} and CC, the fraction bound values are as follows: 76.1% for 2 μ M CC, 89.3% for 5 μ M CC, and 94.4% for 10 μ M CC. Therefore, even at these concentrations, MICAL1^{ACC} is not fully saturated with CC. The fraction bound in this context represents the proportion of available MICAL1^{ACC} that is occupied by CC at a given concentration. This can be calculated using the following formula: $\frac{[(K_d + R_t + L_t) - \sqrt{(K_d + R_t + L_t)^2 - 4R_tL_t}]}{2R_t}$, where R_t is the concentration of MICAL1^{ACC} and L_t is the concentration of CC (Jarmoskaite et al., eLife 2020).

We agree with the reviewer that even near saturation, CC does not inhibit MICAL1^{ACC} to the same extent as full-length MICAL1. We offer a few possible explanations for this observation:

i) CC conformation in solution

Previous crystallographic studies have shown that the CC domain adopts a planar orientation of its three helices. However, our cryoEM structure of full-length MICAL1 revealed that the CC domain can adopt a less planar, axially tilted conformation. The cryoEM structure suggests that for CC to inhibit MICAL activity effectively, it must adopt this axially tilted conformation to engage with the MO domain. Since the purified CC domain in solution does not fully inhibit the MO domain, it suggests that purified the CC domain may not readily adopt the axially tilted conformation in solution. However, it is possible that the CC domain exists in multiple conformations in solution, with slow transitions between them. This equilibrium between different conformations could explain the inability to achieve the same level of inhibition with MICAL1^{ACC} as with full-length MICAL1.

ii) Dissociation constant

The K_D value was determined using a BLI experiment, where the CC domain was immobilized onto a streptavidin-coated sensor. The effect of immobilization on the overall conformation of the CC domain is unknown. If, for example, immobilization artificially stabilizes the axially tilted conformation, the calculated K_D might overestimate the actual K_D , which could explain the lack of full inhibition at the tested concentrations. Additionally, the BLI experiments were conducted in the absence of F-actin, whereas the pyrene-based depolymerization assay was performed in the presence of F-actin. Our study suggests that the binding sites for F-actin and CC partially or fully overlap, indicating competition between CC and F-actin. Therefore, in the presence of F-actin, higher concentrations of CC may be required to achieve effective inhibition than the K_D determined in the absence of F-actin would suggest. We attempted to repeat the BLI experiment with F-actin present, but encountered significant non-specific binding to the reference sensor.

We have incorporated a discussion of these points into the manuscript to address this issue in greater detail. Please see the Results section.

2B) On my first reading, I was struck with how much the authors say that MICAL1 pellets with F-actin (i.e., full-length autoinhibited MICAL1 pellets with F-actin). In particular, the authors say in the results “Although the purified CC domains did not exhibit direct binding to F-actin, approximately $21.8 \pm 1.7\%$ of the full-length MICAL1 was pelleted with actin (Fig. 5 a-c).” My first thought was that this is >20% of the autoinhibited MICAL and thus does not match to their contention that it has little to no activity (Fig. 3a-b and throughout the manuscript). However, after a closer look at comparing their data in Fig. 5a (with F-actin) and Fig. 5b (without F-actin), it is clear that there is pelleting without F-actin. There may also be a little more loaded/darker staining in Fig. 5a. In any case, there is MICAL1 in the pellet without F-actin (Fig. 5b) and so this should be factored into their statement “approximately $21.8 \pm 1.7\%$ of the full-length MICAL1 was pelleted with actin”. Therefore, the labeling of the graph in Fig. 5c needs to be updated – since it is not the same as “pelleted with F-actin”, as said in the results. In short, if the authors want to say something in the text about the pelleting with F-actin they need to subtract out the pelleting on its own and present that in a graph. Moreover, upon comparing the amount of the pellet vs soluble in Fig. 5a, it looks like much less than 20% of MICAL1 is in the pellet compared to the amount in the soluble (and that would be even less when the amount in the pellet in 5b is subtracted). For example, the difference between S and P for the Delta CC (Fig. 5a) seems to

match somewhat to what is presented the graph (Fig. 5c), but the difference between the S and P for MICAL1 does not match.

Response: Thank you for raising this important point. In the original manuscript, we used co-sedimentation assays to demonstrate that the F-actin and CC α 1 helix compete for binding to the MO domain. After ultracentrifugation, we quantified the proteins in both the pellet and supernatant using SDS-PAGE. However, we acknowledge the reviewer's concern regarding the inherent challenges of this approach, including the minor pelleting of MICALs in the absence of F-actin, variability in sample loading on the SDS-PAGE, and also the limitations related to quantification, such as sensitivity, dynamic range, non-linearity of staining, and resolution.

To address these challenges and to provide a more accurate and reliable analysis, we implemented a more robust and sensitive technique: Hydrogen-Deuterium Exchange (HDX) coupled with mass spectrometry. This advanced approach allowed us to confirm that the binding site for the CC α 1 helix on the MO domain partially or fully overlaps with the F-actin binding site, consistent with the findings of our initial co-sedimentation assays.

Given the limitations of the co-sedimentation assay and the superior quality of data obtained through HDX, we have decided to remove the co-sedimentation results from the manuscript and replace them with the more reliable and detailed findings from the HDX experiments.

2C) The manner in which the authors discuss the CC binding and inhibiting the MO does not match the data. In particular, the authors data support (and they state) that the CC does not inhibit the MO (Fig. 3a, 3c, 5a-c) and the CC does not bind to the MO (Fig. 3d). Yet, within the text they contradict their results. For example, the title of Fig. 5 is "F-actin and the CC α 1 helix compete for binding to the MO domain". Likewise, in the legend of Fig 6 they say "Initially, MICAL1 activity is autoinhibited by an intramolecular interaction between the MO domain and the CC α 1 helix. This interaction sterically hinders F-actin from binding to the MO domain." So too, in the results they say "Thus, it appears that the binding site of the CC α 1 helix on the MO domain either fully or partially overlaps with the F-actin binding site." And in the discussion they say "Our co-sedimentation experiments suggest that the binding site of the CC α 1 helix on the MO domain either fully or partially overlaps with the F-actin binding site." Thus, these statements do not match their data and so they need to be revised. Further, the authors results show that the MO is sufficient to bind to F-actin (Fig. 5a) but the MO is not sufficient to bind to the CC (Fig. 3d). Thus, the authors statement that "the binding site of the CC α 1 helix on the MO domain either fully or partially overlaps with the F-actin binding site" is not the accurate. The authors need to rework these statements and any others that give this mixed message. See also Comment #2D.

Response: Thank you for your detailed feedback regarding the discussion of CC binding and its inhibition of the MO domain. We appreciate your careful reading and agree that certain statements and descriptions in our manuscript may have inadvertently conveyed a message that contradicts our findings. This confusion stemmed from the incorrect use of terminology, specifically our unintentional use of the term "CC domain" to describe two distinct conformations with different effects on the MICAL protein.

Specifically, we referred to both the axially tilted conformation of the CC domain, which engages the MO domain and imposes autoinhibition, and the planar conformation, observed in Rab-activated MICAL, which does not engage the MO domain, using the same term. We recognize that this

distinction is crucial for accurately interpreting our results and understanding the role of the CC domain in MICAL regulation. In response, we have carefully reviewed the manuscript to identify and correct any statements that might have conveyed a mixed message.

2D) Elsewhere the authors say “These findings suggest the CC domain alone cannot autoinhibit the MO domain's depolymerization activity, indicating that autoinhibition requires additional elements from the L1-CH-L2-LIM-L3 region.” This statement by the authors is supported by their structure of the full-length MICAL1 in which (as they say in the results) they see interactions between the CH, LIM, and L2 α 1 linker regions and the CC. Indeed, the authors say “These data suggest that the CC domain, not alone but particularly in association with the tripartite CH-L2 α 1-LIM assembly, exhibits a competitive binding interaction with F-actin for the MO domain.”, “In summary, the tripartite assembly of CH-L2 α 1-LIM plays a pivotal role in the binding of the CC domain to the MO domain, enabling the CC domain to adopt a conformation that binds the MO domain and inhibits its catalytic function.”, and “Our findings, supported by biochemical and functional analyses, indicate that MICAL1 autoinhibition hinges on the binding of the CH and LIM domains, facilitated by a helical region of the long linker, to the CC domain. This interaction maintains the CC domain in a specific conformation, enabling it to interact with the catalytic domain and thus enforce autoinhibition.” In light of these statements (and many others that are similar) by the authors, it is important to test for this type of mechanism. In particular, the authors should generate the protein they describe, i.e., containing the CH-LIM-CC domains (i.e., all the regions of MICAL1 that the authors mention that are C-terminal to the MO) and test it for its ability to inhibit the MO. This can be done relatively easily using one of the assays that they authors have already employed here (TIRF microscopy, pyrene-actin, or F-actin sedimentation assays)). This will provide a specific experiment to further test their hypothesis of the mechanism of MICAL1's autoinhibition.

Response: Thank you for your insightful feedback and for highlighting the importance of experimentally testing the proposed mechanism of MICAL1 autoinhibition. We appreciate your suggestion to generate and test a construct containing the CH-LIM-CC domains to assess its ability to inhibit the MO domain. We agree that this experiment would provide valuable insights and serve as a direct test of our hypothesis regarding the role of the CH-LIM-CC assembly in facilitating autoinhibition.

To address reviewer's suggestion, we produced and purified the CH-LIM-CC (MICAL1 Δ MO) construct in insect cells and conducted pyrene-actin depolymerization assays. We observed substantial inhibition of the MO domain across three different molar ratios between the MO domain and the CH-LIM-CC construct. Importantly, the level of inhibition was comparable to that observed with full-length MICAL1, thereby validating our proposed model of autoinhibition. We have included the results of these experiments in a revised version of the manuscript.

3) Related to the mechanism of “activation”. It is clear that the authors have uncovered structural mechanisms related to the autoinhibition of MICAL proteins, but there is little structural information on the activation mechanisms. The authors have formulated their activation model based on modeling of previous structures of fragments of MICAL1 versus having a structure of the full-length autoinhibited MICAL and structure of the full-length MICAL in the presence of Rab to compare or other similar types of structural experiments and experimental testing of those ideas/observations. For example, the authors say based on their modeling “the CC high-affinity binding site for Rab10 in the complex overlaps with the binding site for the tripartite assembly CH-L2 α 1-LIM (Fig. 4c), indicating mutually exclusive binding to the CC domain.” and “the axial tilt of CC α 3 in autoinhibited

MICAL1 precludes low-affinity Rab10 binding (Fig. 4c). Yet, it is not clear how the binding of Rab would displace/override/dissociate the binding of the tripartite assembly and the autoinhibitory interaction with the MO. Indeed, the authors also state their uncertainty regarding other areas related to activation including: “While it remains uncertain whether this results in a complete or partial dissociation of the CC domain from the MO domain, this conformational shift likely exposes the F-actin binding site. Additionally, the conformation of the Rab-bound CC domain may be further stabilized by a second Rab molecule, enhancing the activation process.” Thus, while their modeling provides interesting insights into the mechanism of activation and I don’t believe that further experimentation in this direction is needed for this paper, the authors should scale back their use of activation in the title, abstract, and paper. Perhaps a title of “Structural basis of MICAL autoinhibition with insights into its activation” or something similar.

Response: Thank you for your feedback. We acknowledge the limitations regarding the activation mechanism. As suggested, we have scaled back the emphasis on activation in the title and paper. We have revised the title to: “Structural basis of MICAL autoinhibition” to better reflect the focus of our study.

4) The authors make multiple statements in the text but don’t cite a figure or present a figure to support their statement. It would help the general audience for the authors to include this. For example,

4A) “In the cryoEM structure of MICAL1, the CH domain primarily interacts with the proximal part CC α 2 helix through a combination of hydrophilic and hydrophobic interactions.”  The authors need to refer to a figure/part of a figure or present a more detailed figure to support their statement.

Response: We have created Supplementary Figure S4 to illustrate the interaction between the CH domain and the proximal part of the CC α 2 helix in the cryoEM structure of MICAL1. The manuscript has been updated to reference this new figure.

4B) “Consistent with earlier predictions, the LIM domain (residues 695-757) features two contiguous zinc fingers, separated by a Phe-Arg pair. The LIM domain engages the central part of the CC α 1 and CC α 2 mainly through hydrophobic contacts. It also binds to the CH domain, with hydrophobic interactions between Phe716 from LIM and Val585 and Val586 from CH, further cemented by a salt bridge (Asp596 - Arg709). Intriguingly, the LIM domain also forms van der Waals contacts with the region between MO β 11 and MO α 11, crucial for the stability of the MO-CC domain interaction.”  The authors need to refer to a figure/part of a figure or present a more detailed figure to support their different statements here.

Response: We have prepared Supplementary Figure S4 to provide detailed visual support for the interactions described between the LIM domain and other regions. This figure highlights the hydrophobic contacts, salt bridge, and van der Waals interactions. The manuscript has been updated to reference this new figure.

4C) “In the cryoEM structure, the CH domain binds to the CC domain.”  The authors need to refer to a figure/part of a figure or present a more detailed figure to support their statement.

Response: We have created Supplementary Figure S4 to show the binding interaction between the CH domain and the CC domain in the cryoEM structure. The manuscript has been updated to

reference this new figure.

5) Some additional labeling changes/updates should be made to the figures and text:

5A) In Fig. 1b  label each of the CC helices

Response: Thank you for this suggestion. We have updated Figure 1b to label each of the CC helices as requested.

5B) The authors say in the results “Surprisingly, the addition of the purified CC domain at varying concentrations failed to inhibit depolymerization (Fig. 3a, c)”  but looking at the context it should be modified to “.....the addition of the purified CC domain at varying concentrations to the MO failed to inhibit depolymerization.....”

Response: We appreciate this clarification. We have revised the text to “.....the addition of the purified CC domain at varying concentrations to the MO failed to inhibit depolymerization.....” in the Results section for improved accuracy and clarity.

5C) For Fig. 5c: it should indicate on the figure that is with F-actin. For Fig. 5A and 5B, it would also be good to indicate on the figure that it was with F-actin and without F-actin, respectively

Response: In light of the limitations associated with the co-sedimentation assay, we have decided to remove the co-sedimentation results originally presented in Fig. 5. These results have been replaced with more robust and reliable data obtained from Hydrogen-Deuterium Exchange (HDX) experiments. For further details, please see our response to point 2B.

6) Statements/sentences in the manuscript need to cite references (so that the reader is not left wondering what is the basis of these statements). For example,

6A) “Similar to the previously published CC structures,”  missing the citations (refs 26 and 27)

Response: We apologize for the oversight. The sentence has been updated to include the missing citations (refs 26 and 27).

6B) “This domain is highly similar to that of the CH domain of human MICAL1 determined via NMR.” -> missing the citation (Solution structure of calponin homology domain of Human MICAL-1. Hongbin Sun, Haiming Dai, Jiahai Zhang, Xianju Jin, Shangmin Xiong, Jian Xu, Jihui Wu & Yunyu Shi; Journal of Biomolecular NMR (2006) 36:295–300)

Response: We have added the appropriate citation to the manuscript.

Reviewer #3 (Remarks to the Author):

The present manuscript delves into an investigation of the molecular mechanisms underlying the regulation of MICAL1, a protein crucial for cellular dynamics through its role in actin filament disassembly. By integrating cryo-EM structural analysis, alongside biochemical and functional assays, the authors provide comprehensive insights into MICAL1's autoinhibition and activation mechanisms.

The study reveals the cryo-EM structure of human MICAL1 at a nominal resolution of 3.1 Å, highlighting that autoinhibition is mediated by an intramolecular interaction between its N-terminal catalytic and C-terminal coiled-coil domains, which obstructs F-actin interaction. Moreover, the authors elucidate that allosteric changes in the coiled-coil domain, coupled with the binding of the CH-L2 α 1-LIM domains to this coiled-coil region, are critical for MICAL1 activation and autoinhibition. Furthermore, the findings suggest that these regulatory mechanisms are evolutionarily conserved, indicating a potential universality across the MICAL protein family. This study significantly advances the understanding of MICAL1 regulation and provides a foundational basis for future research into its role in cellular dynamics and potential implications in disease contexts where actin dynamics are disrupted.

Comments:

1. The authors utilized AlphaFold to model the linker regions and conducted a 100 ns molecular dynamics simulation. However, a more in-depth discussion on the reliability and accuracy of these models is necessary. Have additional validations been performed, such as comparisons with experimental data or verification through other computational methods? These validations would help strengthen the interpretation of the models and the credibility of the results.

Response: Thank you for your insightful comments and suggestions. We appreciate your attention to the details of our modeling approach.

In the original manuscript, we utilized the high quality and resolution of our cryoEM map to confidently model the four main domains of MICAL1: MO, CH, LIM, and CC. However, the cryo-EM density for the long linker regions connecting these domains was sparse, indicating their inherent flexibility, which prevented us from modeling these regions. Nonetheless, we observed additional strong cryoEM density in areas close to, but not directly corresponding with, the main domains, suggesting that these densities might originate from the linker regions.

To better understand these extra densities, we used AlphaFold modeling followed by molecular dynamics (MD) simulations. The AlphaFold model for human MICAL1, available in the AlphaFold Protein Structure Database (ID: Q8TDZ2) and generated using the AlphaFold Monomer v2.0 pipeline, did not align well with our experimental model. Specifically, there were discrepancies in the positions of individual domains, the topology of secondary structures, and the geometry and pitch angles of the CC domain. Consequently, we fixed the positions of the main domains in our experimental model and used AlphaFold specifically to model the linker regions, which we then subjected to MD simulations.

This approach allowed us to discern one of the extra densities as likely corresponding to a helical structure within the L2 linker, which we termed the L2 α 1 helix. Given the good quality of the cryo-EM density in this region, we were able to confidently model this helix, confirming that our prediction matched the experimental data.

In response to your comment, we realized that our explanation of this approach was not sufficiently detailed in the original manuscript. We have now expanded the Results section to provide a clearer and more thorough description.

Additionally, we observed extra density near the proximal part of CC α 1, although it was not as well-defined as the L2 α 1 helix. This bulk region likely represents an arrangement of small helices, with a

narrow density extending from it that suggests the presence of a peptide chain. While AlphaFold predicted this region to be a proline-rich region, the local low resolution made it difficult to build this segment unambiguously or confirm it experimentally. We have included images of the cryo-EM density for this region in the Supporting Material – Fig. S5.

To address your concerns about the reliability and accuracy of the models generated using AlphaFold and subsequent MD simulations, we have included pLDDT and PAE scores in the revised manuscript. Given the dynamic nature of the linker regions, direct experimental validation remains challenging, and we believe that exploring this further would extend beyond the scope of our current study. Therefore, we have not pursued additional experimental validation of the predicted linker regions, except for the L2 α 1 helix.

2. The authors used AlphaFold for modeling the linker regions and conducted a 100 ns molecular dynamics simulation, describing the interactions between the four domains and the linker regions. It is recommended that the authors include illustrations to depict and describe the movement patterns and correlations of each domain and the linker regions.

Response: We appreciate your suggestion to include illustrations that depict the movement patterns and correlations between the domains and linker regions. In response to your recommendation, we have created and added a series of illustrations to the Supplementary material of the revised manuscript. These illustrations visually represent the movement patterns of the four domains (MO, CH, LIM, and CC) and the linker regions throughout the 100 ns molecular dynamics simulation.

3. In Figure S3(b), RMSF of C α atoms (orange) with standard deviation depicted in grey. I believe this should be RMSD. Please verify.

Response: Thank you for bringing this to our attention. We have carefully reviewed the figure legend and acknowledge the oversight. We have corrected the figure legend to accurately state: "RMSD of C α atoms (orange) with standard deviation depicted in grey."

Additionally, we have revised the original Figure S3(b) by recalculating the RMSD values using the experimental cryoEM structure as the reference. The updated figure now reflects these recalculations, ensuring that it aligns with the intended analysis.

4. The authors mention that the linker regions exhibit significant flexibility and explore a wide conformational space, which plays a crucial role in the function and structure of MICAL1. It would be beneficial if the authors could provide a detailed description of the different conformations of the linker regions observed during the simulation and identify which conformations are particularly relevant to biological functions. Additionally, the authors noted that certain segments exhibit relative rigidity, suggesting an association with the core structure. Please explain the reasoning behind this association with the core.

Response: Thank you for your thoughtful comments and suggestions. In our manuscript, we discussed the significant flexibility of the linker regions and their ability to explore a wide range of conformations during the simulations. We observed that these linker regions can adopt several distinct conformations, from extended, flexible structures to more compact, folded arrangements. To address your request for more detail, we have now included images in the revised manuscript that

illustrate the different conformations adopted by each linker region (L1, L2, and L3) throughout the simulation.

Regarding the segments of the linker regions that exhibit relative rigidity, our original manuscript primarily focused on the L2 α 1 helix. This helix is crucial for maintaining the axial tilt of CC α 3 in the autoinhibited state, which is essential for the CC domain to engage with the MO domain and impose autoinhibition. The rigidity of the L2 α 1 helix is likely due to its proximity to and interactions with the CH, LIM, and CC domains. This helix likely serves as a stabilizing anchor, preserving the structural integrity of the tripartite CH-L2 α 1-LIM assembly. To demonstrate the rigidity of the L2 α 1 helix during simulations, we have analyzed its interactions with the CH, LIM, and CC domains and calculated the occupancy of these interactions throughout the simulations. We have expanded our discussion in the revised manuscript to explain the significance of this association with the core structure and have included a table of these occupancy calculations in the Supplementary Materials - Fig. S5.

5. The simulations describe the MO, CH, LIM, and CC domains as relatively rigid entities, with some highly flexible local regions, particularly in the loop region connecting the helices of the CC domain. explain why the loop region connecting the helices of the CC domain exhibits higher flexibility.

Response: Thank you for the opportunity to clarify the dynamics observed in our simulations. The loop regions connecting the helices of the CC domain exhibit higher flexibility due to their inherent structural characteristics. Unlike the helical regions, which are stabilized by extensive intra-helix hydrogen bonding and hydrophobic interactions, the loop region typically lacks these stabilizing forces, resulting in increased conformational freedom. This increased flexibility is a common feature in loop regions across many proteins, where it plays a crucial role in facilitating dynamic processes such as conformational changes, protein-protein interactions, or accommodating binding partners.

In the case of the CC domain of human MICAL1, this flexibility is particularly significant. Previous crystallographic studies have shown that the CC domain primarily adopts a planar orientation of its three helices. However, our cryoEM structure of the full-length MICAL1 revealed that the CC domain can also adopt a less planar, axially tilted conformation. The flexibility of the loop region appears to be key in allowing these adjustments in the relative positioning of the helices, which is necessary for the transition between the autoinhibited and activated states of the protein. We have expanded the discussion in the manuscript to include this explanation, highlighting the structural and functional significance of the loop region's flexibility within the CC domain.

6. The authors state that the binding of Rab10 to MICAL1 results in the dissociation of the CH-L2 α 1-LIM complex. Please elaborate on how this dissociation process is regulated and clarify whether it is a direct or indirect result.

Response: Thank you for your insightful comment. In our original manuscript, we proposed a theoretical mechanism for MICAL1 activation by Rab10 molecules. Drawing on our cryoEM structure and previous crystallographic studies of the CC-Rab complex, we suggested that the first Rab10 molecule binds to a site on the CC domain that, in the autoinhibited form of MICAL1, is occupied by the tripartite CH-L2 α 1-LIM assembly. This binding event triggers the dissociation of the CH-L2 α 1-LIM complex from the CC domain. The resulting absence of the L2 α 1 helix, which stabilizes the axially tilted conformation of CC α 3 in the autoinhibited state, allows CC α 3 to straighten, transitioning the

CC domain to a more planar arrangement. This conformational shift then facilitates the binding of a second Rab10 molecule to a secondary site, further stabilizing the planar conformation.

In light of your comment, we considered the possibility that the order of Rab10 binding could be reversed. In this alternative scenario, the first Rab10 molecule could bind to the secondary site first, inducing the straightening of the CC domain and leading to the dissociation of the tripartite CH-L2 α 1-LIM assembly. This would then allow a second Rab10 molecule to bind to the initial site. Both binding sequences are theoretically plausible. However, we currently lack experimental evidence to definitively support either hypothesis. Furthermore, we do not yet have detailed experimental data on the regulation of the tripartite assembly's dissociation. Addressing this would likely require further studies, potentially including the structural determination of MICAL1 in complex with Rab molecules.

In the revised manuscript, we have expanded the discussion to include the possibility of Rab binding in the opposite order. We also recognized that we inadvertently switched the terms for the low- and high-affinity binding sites. We apologize for this oversight, and we have corrected it in the revised manuscript.

7. Regarding how the conformation of the Rab-bound CC domain further stabilizes the activation process, could the authors provide more details or a theoretical explanation?

Response: Thank you for your comment. In our previous response, we explained that when Rab10 binds to the initially available site on the CC domain, it triggers the straightening of the CC α 3 helix, transitioning the CC domain from an axially tilted to a more planar and active conformation. Once the CC domain adopts this planar conformation, a second Rab10 molecule can bind to a newly exposed site on the CC domain. The binding of this second Rab10 molecule likely plays a crucial role in further stabilizing the active conformation of the CC domain. The dual Rab binding may create a more rigid and stable structural framework, which could prevent the CC domain from reverting to its autoinhibited state. This stabilization ensures that MICAL1 remains in its active conformation, promoting downstream signaling processes. Theoretically, this mechanism of stabilization can be explained by allosteric effects, where the binding of Rab10 induces a conformational shift that is energetically favorable, locking the CC domain in an active state. In the revised manuscript, we have expanded the discussion to include this theoretical explanation.

Title: Structural basis of MICAL autoinhibition

Tracking number: NCOMMS-24-22666

Authors: Matej Horvath, Adam Schrofel, Karolina Kowalska, Jan Sabo, Jonas Vlasak, Farahdokht Nourisanami, Margarita Sobol, Daniel Pinkas, Krystof Knapp, Nicola Koupilova, Jiri Novacek, Vaclav Veverka, Zdenek Lansky and Daniel Rozbesky

October 1, 2024

Dear Reviewers,

Thank you very much to all the reviewers for their valuable feedback and thoughtful comments. We are grateful for your time and effort in reviewing our manuscript. Your insights and suggestions have helped improve the clarity and quality of our work, and we have made revisions in line with your recommendations.

REVIEWER COMMENTS

Reviewer #1 (Remarks to the Author):

The authors have satisfactorily addressed the criticism. I believe the revised MS is acceptable for publication NC

Response: Thank you for your positive feedback. We are pleased to hear that the revisions have addressed your concerns.

Reviewer #2 (Remarks to the Author):

The authors have done a nice job of revising the manuscript, including conducting critical functional assays (Fig. 3f) and providing clearer insights into the questions addressed in Fig. 5. However, the authors inadequately addressed a few of my initial comments and thus still need to address a couple of their findings in the paper.

Response: We appreciate your detailed and constructive feedback. Below, we have carefully revisited the specific points that were previously inadequate, and we provide a more thorough explanation and clarification for these findings. We hope that these additional revisions now satisfactorily address the concerns raised.

1) The major issue that I mentioned in the first version of this manuscript is the relatively poor job that the CC alone does in inhibiting the Delta CC construct. However, instead of adequately responding to my comments and addressing this in the manuscript, the authors focused on addressing the differences between the TIRFM and pyrene assays, which were already understood and not the point I was getting at. Therefore, the manuscript still needs work in this area to accurately reflect the authors data.

1A) The relatively weak inhibition seen in the TIRFM images in Fig. S6 (e.g., compare MICAL1 Δ CC + CC (100 μ M) with MICAL1 Δ CC and also to MICAL1, which is ~fully inhibited) roughly matches the weak inhibition of the CC on the Delta CC (Fig. 3e, compare light red to dark red and also to black, which is ~fully inhibited). Thus, given that these are different assays, I think the relative inhibition in images of the TIRFM data (Fig. S6) is comparable to the relative inhibition of the pyrene actin assays (Fig. 3e). However, instead of the authors seeing the point of my questions (i.e., that the CC is a weak and not a robust inhibitor of the Delta CC), they unnecessarily discussed the differences between the TIRFM and Pyrene actin assays as well as other methodological reasons why this is the case. Of course, these two assays are different but they have been used complimentary for hundreds/thousands of papers over the years. Instead, the authors should follow their data, which say that the CC is not a strong inhibitor of the Delta CC and discuss why that might be the case. Given the other robust data that the authors have now provided (Fig. 3f), it seems likely that once the autoinhibition of MICAL1 is relieved, the CC is not robustly/fully able to inhibit MICAL1's activity again? Thus, as I specifically detail below, this section of the manuscript needs to be reworked to focus on the point that the authors experiments reveal vs simply pointing out differences in methodologies.

Response: We thank the reviewer for their valuable insight. We now fully understand that the primary point being raised is that our data show the CC domain is a weak inhibitor of MICAL1 Δ CC, and this is consistent across both the TIRFM images and the pyrene actin assays. We acknowledge that our previous response focused too much on methodological differences instead of addressing this core observation. In response, we have revised the manuscript to directly address the relatively weak inhibition observed and discuss potential reasons for this. Additionally, please refer to our response to your comments 1E and 1F.

1B) Lines 196-197: Notably, the addition of the purified CC domain to MICAL1 Δ CC led to a marked inhibition of F-actin depolymerization (Fig. 3a, e).  I would recommend either removing the term "marked" or changing it to "noticeable". Having done a number of TIRFM and pyrene assays, I would not say it is marked inhibition but it is convincing. And coupled with Fig. 3f, it convinces the reader that the authors are on the right track. Referring to the actual data in Fig. S6 should also be done in this sentence (Figs. 3a, 3e, S6)

Response: We appreciate the reviewer's careful assessment of our wording and the suggestion to modify the description of the inhibition observed in the assays. We agree that "noticeable" is a more accurate term and have revised the sentence accordingly. We have also included the reference to Fig. S6 as suggested to provide further support for this finding.

1C) The authors helpfully changed the manuscript to use a ratio in Fig. 3 but did not make the change to ratio in Fig. S6. This should be done in Fig. S6 so it is easier for the reader to go between the two figures.

Response: Thank you for pointing out the inconsistency with Fig. S6. We have now updated Fig. S6 to present the data in a ratio format, making it easier to compare with Fig. 3.

1D) Lines 197-200: "Data from the pyrene-labeled actin depolymerization assays (Fig. 3e) were consistent with the trends observed in the single actin filament TIRF microscopy (Fig. 3a), though the degree of inhibition observed in the pyrene assays was less pronounced, likely due to inherent methodological differences between the two techniques." The authors should remove "likely due to inherent methodological differences between the two techniques" since they don't know that. Or replace "likely" with "possibly". However, as

mentioned in 1A), I don't see much relative difference in inhibition between the two assays so they could alternatively remove “, though the degree of inhibition observed in the pyrene assays was less pronounced, likely due to inherent methodological differences between the two techniques.”. The authors also need to refer to Fig. S6 (the actual images) anytime they refer to Fig. 3a in the results.

Response: We have revised the sentence in lines 197-200, as suggested, removing the phrase about "inherent methodological differences". We have also added references to Fig. S6 where appropriate.

1E) Lines 200-204: “Interestingly, even at CC domain concentrations near saturation (94.4% bound at 10 μ M, as calculated based on the determined KD), the purified CC domain did not inhibit MICAL1 Δ CC to the same extent as full-length MICAL1. This discrepancy could be attributed to multiple CC conformations in solution or an overestimation of the KD in the BLI experiment due to the absence of F-actin.” I would suggest that there are also other explanations that the authors might want to touch on here or elsewhere (see point 1F).

Response: We thank the reviewer for this thoughtful comment. In response, we have expanded our discussion to address why the CC domain does not fully inhibit MICAL1 Δ CC and why the CC domain alone fails to inhibit the MO domain. This explanation is based on potential conformational differences in the CC domain. Please refer to the last paragraph of the section: *Binding of the CH-L2 α 1-LIM assembly to the CC domain is crucial for MICAL autoinhibition*

While we agree that there are additional potential explanations for this observation, we respectfully acknowledge that we do not have experimental data to support further speculations at this point. Therefore, we have limited our discussion to the conformational aspects of the CC domain and its impact on MICAL1 inhibition.

1F) The data in Fig. 3f are robust; thereby helping to validate the authors proposed model of autoinhibition. Thus, it would seem that there are likely other reasons why the CC does not robustly/fully inhibit the Delta CC vs what the authors mention in Lines 200-204. The authors should at least discuss this as I allude to in 1A) including perhaps that without being connected to more N-terminal regions of MICAL, the CC alone does not adopt the right conformation to robustly/fully inhibit the MO, and/or once the autoinhibition of MICAL is relieved, the CC is not sufficient to robustly/fully inhibit it again, and/or perhaps other reasons. I think this is important to discuss vs simply stating it is due to issues with overestimation of KD, CC solution conformations, or F-actin assays, which is clearly not the case for what they are observing in the pyrene actin assays with MO + delta MO compared to Delta CC + CC.

Response: Please refer to our response to the previous point.

1G) Lines 731-733: “(b, c, e) Data from pyrene-labeled actin depolymerization assays demonstrate overall trends that are consistent with the findings from TIRF microscopy (a). However, the degree of inhibition observed is less pronounced, likely due to inherent differences between the two methods.” This is not an accurate statement: the MO depolymerization is very pronounced using both assays and the inhibition of the MO by the Delta MO is very pronounced in the pyrene actin assays (much more so than the CC inhibition of the Delta CC). So too, same points as above, TIRFM and pyrene actin assays

have been used as complimentary approaches for years, so it is not necessary to mention inherent differences between the methods. Therefore, I would remove the statement “However, the degree of inhibition observed is less pronounced, likely due to inherent differences between the two methods.”

Response: We acknowledge that our initial statement regarding the comparison between the TIRFM and pyrene actin assays may not have been entirely accurate. In light of this, we have revised the statement to remove the mention of less pronounced inhibition and inherent differences between the methods.

Minor:

2) Line 104: “A key feature of the MO-CC interaction involves a region between MO β 17 and MO α 15...”  should be modified to include Fig. S1. So that the reader knows where to access these data: “A key feature of the MO-CC interaction involves a region between MO β 17 and MO α 15 (Figs. 1d, S1)...”

Response: We have added a reference to Fig. S1 in line 104, as suggested.

3) A reference to Fig. S1 should also be added to Lines 107-110 so that it reads “....and Ile931 from the CC α 1 and CC α 2 helices (Figs. 1e, S1)”

Response: We have included a reference to Fig. 1a at the end of lines 135-138 to guide the reader.

4) Lines 135-138: “The shortest linker, L1 135 (residues 490-507), serves as a bridge between the MO and CH domains. The intermediate L2 linker 136 (residues 613-694) connects the CH domain to the LIM domain, and the longest, L3 (residues 758 to 137 907), links the LIM to the CC domain”  Refer to Fig. 1a at the end of this sentence

Response: We have included a reference to Fig. 1a at the end of lines 135-138 to guide the reader.

5) In the legend of Fig. S4c the authors use cyan to indicate a region in the MO. However, they also use Cyan to indicate the L1 region in Fig. 2a. So, I would recommend changing the color of the cyan in Fig. S4c (so it is not misunderstood that they are referring to a region of the L1, which I originally thought perhaps they were).

Response: In response to your suggestion, we have changed the color used in Fig. S4c to avoid confusion with the L1 region depicted in Fig. 2a.

6) For ease to help the reader, please reorder Fig. 3a so that is matches the description of the constructs in the legend from top to bottom (not bottom to top)

Response: We have reordered the panels in Fig. 3a to match the order described in the legend for clarity.

7) Please note a few typos such as: Line 813 “(c) Visualization of the of MO domain with CC α 1 Helix.”  change to “Visualization of the MO domain with CC α 1 Helix.”

Response: We have corrected the typo on line 813, as well as a few other minor errors throughout the manuscript.

Reviewer #3 (Remarks to the Author):

I think the manuscript is now ready for the publish on Nature Communications.

Response: Thank you for your positive comments and for recommending our manuscript for publication.

REVIEWERS' COMMENTS

Reviewer #2 (Remarks to the Author)

The authors have carefully addressed my remaining concerns. I believe the paper is ready for publication

A POINT-BY-POINT RESPONSE TO THIS COMMENT

Reviewer #2 (Remarks to the Author)

The authors have carefully addressed my remaining concerns. I believe the paper is ready for publication

Response: Thank you for your positive feedback. We are pleased to hear that the revisions have addressed your concerns.